# Multigene Phylogeny Reveals Endophytic Xylariales Novelties from *Dendrobium* Species from Southwestern China and Northern Thailand

**DOI:** 10.3390/jof8030248

**Published:** 2022-02-28

**Authors:** Xiaoya Ma, Putarak Chomnunti, Mingkwan Doilom, Dinushani Anupama Daranagama, Jichuan Kang

**Affiliations:** 1Engineering and Research Center for Southwest Biopharmaceutical Resource of National Education Ministry of China, Guizhou University, Guiyang 550025, China; maxy.dela@gmail.com; 2Center of Excellence in Fungal Research, Mae Fah Luang University, Chiang Rai 57100, Thailand; 3School of Science, Mae Fah Luang University, Chiang Rai 57100, Thailand; putarak.cho@mfu.ac.th; 4Innovative Institute for Plant Health, Zhongkai University of Agriculture and Engineering, Guangzhou 510225, China; j_hammochi@hotmail.com; 5Research Center of Microbial Diversity and Sustainable Utilization, Faculty of Science, Chiang Mai University, Chiang Mai 50200, Thailand; 6Department of Biology, Faculty of Science, Chiang Mai University, Chiang Mai 50200, Thailand; 7Department of Plant and Molecular Biology, Faculty of Science, University of Kelaniya, Colombo 11300, Sri Lanka; anubs206@gmail.com

**Keywords:** endophytes, multi-locus phylogeny, orchids, oat media, Xylariomycetidae

## Abstract

Xylariales are common endophytes of *Dendrobium*. However, xylarialean species resolution remains difficult without sequence data and poor sporulation on artificial media and asexual descriptions for only several species and old type material. The surface-sterilized and morph-molecular methods were used for fungal isolation and identification. A total of forty-seven strains were identified as twenty-three species belonging to Apiosporaceae, Hypoxylaceae, Induratiaceae, and Xylariaceae. Five new species—*Annulohypoxylon moniliformis*, *Apiospora dendrobii*, *Hypoxylon endophyticum*, *H. officinalis* and *Nemania dendrobii* were discovered. Three tentative new species were speculated in *Xylaria*. Thirteen known fungal species from *Hypoxylon*, *Nemania*, *Nigrospora*, and *Xylaria* were also identified. Another two strains were only identified at the genus and family level (*Induratia* sp., *Hypoxylaceae* sp.). This study recorded 12 new hosts for xylarialean endophytes. This is the first report of Xylariales species as endophytes from *Dendrobium aurantiacum* var. *denneanum*, *D. cariniferum*, *D. harveyanum*, *D. hercoglossum*, *D. moniliforme*, and *D. moschatum*. *Dendrobium* is associated with abundant xylarialean taxa, especially species of *Hypoxylon* and *Xylaria*. We recommend the use of oat agar with low concentrations to induce sporulation of *Xylaria* strains.

## 1. Introduction

*Dendrobium* Sw. is one of the three largest genera in Orchidaceae [1]. Many *Dendrobium* orchids have important medicinal and ornamental values [2,3]. However, the majority of *Dendrobium* species are endangered due to low germination rates, habitat destruction, and over-exploitation as reported in the IUCN (International Union for Conservation of Nature) Red List of Threatened Species. Fungal endophytes play an important role in orchid development and defense against stress [4,5,6,7]. The symbiotic germination of eleven orchid species can be enhanced by some fungal endophytes [8,9]. Extracts of both *Dendrobium* and fungal endophytes have been found to possess various bioactivities such as angiogenesis inhibitory, anti-cancer, anti-inflammatory, anti-mutagenic, and anti-oxidative bioactive properties [10,11,12,13,14,15,16].

Xylariales were introduced by Nannfeldt in 1932 as a large order in Sordariomycetes of perithecial Ascomycota with eight-spored unitunicate asci, usually with an apical J+ apparatus and ellipsoidal, dark ascospores [17,18]. Fifteen families and 44 accepted genera have been recorded in this order [18,19,20,21]. The xylariaceous taxa have a polyphyletic topology in their phylogeny, and many old xylarialean species lack sequence data or require further verification with herbarium-defined types and repeated collections [17,19,22,23,24]. They are common endophytes and existed as hemibiotrophs, necrotrophs, pathogenes, and saprobes that occur on monocotyledons, dicotyledons, dung, and even in some arthropod animals with diverse lifestyles in both terrestrial and aquatic environments [18,23,25,26,27,28]. Xylariales are the best-investigated filamentous fungal group due to their wide distribution and production of bioactive secondary metabolites that possess various bioactivities [29,30,31,32,33]. Xylariaceae is one of the largest families of Xylariales, which was introduced by Tulasne and Tulasne in 1863 [20,34]. Hyde et al. (2020) verified the Xylariaceae placement in the subclass Xylariomycetidae and estimated the subclass divergence time at 278 Mya.

*Dendrobium* species contain a large community of xylariaceous endophytes [35,36,37]. Several xylariaceous genera, viz. *Hypoxylon*, *Nemania*, and *Xylaria* have been isolated from *Dendrobium* orchids in tropical and subtropical areas [35,37,38,39]. However, few reports focus on species resolution, and most were carried out with single gene-based phylogeny [35,36,40,41]. A single gene is inadequate for xylariaceous species delimitation [18,20,23,42,43]. A considerable number of sterile mycelia from endophytic isolates make accurate identification difficult, particularly the typical genera *Xylaria*, and morphology on artificial media is rare [44,45,46,47,48,49]. Therefore, a multi-gene phylogenetic approach is recommended while it is also encouraged to induce sporulation for better species resolution [26,42].

In the present study, we investigated Xylariales from 23 *Dendrobium* samples (including eight unidentified *Dendrobium* species) collected from southwestern China and northern Thailand. *Dendrobium cariniferum* Rchb. f., *D. chrysotoxum* Lindl., *D. fimbriatum* Hook., *D. harveyanum* Rchb.f., *D. hercoglossum* Rchb. f., *D. loddigesii* Rolfe, *D. moniliforme* (L.) SW, *D. moschatum* (Buch-Ham.) SW, and *D. primulinum* Lindl. are native species to Southeastern Asia (http://www.orchidspecies.com/, 3 July 2021), while *Dendrobium officinale* Kimura & Migo (Tie Pi Shi Hu in China) is endemic to China [50]. In this study, we provided descriptions and illustrations of 47 endophytic xylariaceous strains belonging to Apiosporaceae, Hypoxylaceae, Induratiaceae, and Xylariaceae. The concatenated sequence data of ITS-LSU-TUB2-TEF-1α and ITS-LSU-TUB2-RPB2 were used for phylogenetic species analysis in Apiosporaceae and Hypoxylaceae/Xylariaceae, respectively.

## 2. Materials and Methods

### 2.1. Sample Collection

Healthy leaves, roots, and stems of 23 *Dendrobium* samples (with eight unidentified species) were collected from six different collecting sites in southwestern China and northern Thailand (Table 1). Materials were packed in zip-lock bags or tubes containing silica gel on ice. Fungal isolation was carried out within 48 h following collection.

### 2.2. Fungal Isolation and Cultivation

Surface sterilization of the orchid tissues and endophytic fungal isolation was carried out as described by Nontachaiyapoom et al. (2010) with modifications [51]. The tissues were first washed with running water. Leaves, roots, and stems were then soaked in a solution containing 3% (*v*/*v*) H_2_O_2_ and 70% (*v*/*v*) ethanol for 5 min, and then rinsed with sterile distilled water three times. Sterilized tissues were cut into 2 mm^2^ pieces and put on potato dextrose agar (PDA) containing 50 μg/mL oxytetracycline, 50 μg/mL penicillin, and 50 μg/mL streptomycin [52]. The surface sterilization was tested via the imprinting method as described in Petrini (1991) and Hyde (2008) [53,54]. Samples were incubated at 28 °C in the dark. Vegetative mycelia were cut and transferred to fresh PDA to obtain pure cultures. The growth rates were evaluated by measuring the colony diameter and growth time. The pure cultures were deposited at China General Microbiological Culture Collection Center (CGMCC), the Culture collection of Guizhou Agricultural College, Guizhou University (GZAC), and Mae Fah Luang University Culture Collection (MFLUCC).

### 2.3. DNA Extraction and Amplification

DNA samples were extracted from fresh mycelia scraped from pure fungal cultures using the EZgeneTM Fungal gDNA Kit (GD2416, Biomiga, San Diego, CA, USA) following the manufacturer’s protocol. The amplification reactions were performed using reagents purchased from BIOMIGA (San Diego, CA, USA). Each 25 μL amplification reaction contained 12.5 μL of 2*Bench TopTM Taq Master Mix (0.05 units/μL Taq DNA polymerase, 0.4 mM dNTPs and 4 mM MgCl_2_); 2 μL of forward and reverse primers; 1 μL of DNA template and 9.5 μL of threefold-distilled water. Five loci were selected for DNA amplification including ca. 900 bp section of the internal transcribed spacer (ITS), ca. 1k bp segment of large subunit (LSU), ca. 1.2k bp fragment of RNA polymerase II core subunit (RPB2), ca. 250 bp fraction of the elongation factor 1-alpha (TEF-1α), and ca. 0.5k bp fragment of β-tubulin (TUB2) [23,55]. The primers used in this study and PCR thermal cycling conditions are listed in Table 2. The PCR products were viewed on 1% agarose gel electrophoresis stained by 4S green nucleic acid (Sangon Biotech Co., Ltd., Shanghai, China) to check the quality and then sent to Sangon Biotech Co., Ltd. (Shanghai, China) for purification and sequencing.

### 2.4. Phylogenetic Analysis

The original DNA sequence data were manually trimmed and assembled into contigs using ContigExpress (Vector NTI suite 6.0, Informax). The consensus sequences were subjected to BLASTn in the NCBI GenBank database for the initial screening of the most similar sequences, particularly for those of ex-type/ex-epitype strains (www.ncbi.nlm.nih.gov/BLAST, 3 July 2021). All new sequences in this study were deposited in GenBank (Appendix A).

The selected sequences were aligned using MAFFT version 6 [56] (http://mafft.cbrc.jp/alignment/server/, 3 July 2021). Aligned datasets were visually inspected and misaligned regions were trimmed by TrimAL v. 1.2 in PhyloSuite v. 1.2.2 [57,58] and AliView [59]. All base pair differences, excluding gaps, were calculated by MEGA 7.0 [60]. Gaps were treated as missing data in maximum likelihood and Bayesian inference trees. For the Apiosporaceae tree, the general time-reversible (GTR) model with the gamma distribution option was implemented for the TEF-1α and TUB2 datasets. The GTR model with Inv-Gamma distribution was selected for the ITS and LSU datasets. For another tree, the GTR model with the Inv-Gamma distribution option of nucleotide substitution was selected for the ITS, LSU, and RPB2 datasets. The Hasegawa, Kishino & Yano (HKY) model with Inv-Gamma distribution was used for the TUB2 datasets. The ML tree was constructed using RAxML-HPC2 on XSEDE in CIPRES Science Gateway V. 3.3 (http://www.phylo.org/index.php/, 4 July 2021). The BI tree was achieved by MrBayes version 3.2.7 via the public resource platform CIPRES Science Gateway V3.3 (Ronquist & Huelsenbeck 2003, https://www.phylo.org/, 4 July 2021). The concatenated dataset was partitioned, and the ultrafast bootstrap [61] implemented in the IQ-TREE software [62] was used to estimate the best fitting models according to the Bayesian information criterion (BIC).

Two sets of six simultaneous independent chains of Markov Chains Monte Carlo (MCMC) simulations were run for 5,000,000 (For Apiosporaceae phylogenetic tree) and 400,000,000 (For Xylariaceae related taxa phylogenetic tree) generations, 25% of trees were discarded, and the remaining trees were used to calculate the posterior probabilities. Convergence was assumed when the standard deviation of split sequences was less than 0.01. The best scoring RAxML trees with bootstrap values of maximum likelihood (ML) (equal to or above 70%) and Bayesian posterior possibilities (BPP) from MCMC analysis trees (equal to or above 90%) are shown in phylogenetic trees. The two phylogenetic trees were submitted to TreeBase (http://purl.org/phylo/treebase/phylows/study/TB2:S28326; http://purl.org/phylo/treebase/phylows/study/TB2:S28327, 5 July 2021). All trees were viewed in FigTree v.1.4.3 (University of Edinburgh, UK) and edited in Adobe Illustrator v. 25.2. (Adobe Inc., Mountain View, CA, USA)

### 2.5. Morphological Study

Pure strains were cultivated on 2% malt extract agar (MEA). Fresh mycelia were transferred to potato dextrose agar (PDA), 4% oat agar (OA), and water agar (WA) using sterilized toothpicks for sporulation [45]. Cultures were kept in a dark cabinet at room temperature (28 °C) and observed every seven days or less. The growth rate was evaluated when fungal mycelia nearly covered the whole plate. Fungal structures were observed and captured with a stereomicroscope (SteREO Discovery. V8, Carl Zeiss Microscopy GmBH, Jena, Germany). Cross-sections of conidiomata, or ascomata-like, structures were mounted in 10% hydrogen peroxide and observed using a Nikon ECLIPSE 80i compound microscope with a Cannon 600D digital camera (Japan). The size of the morphological characteristics was measured with Tarosoft (R) Image Frame Work (IFW) v.0.9.7 and the photo plates were made by Adobe Photoshop CS 22.2.0 (Adobe Inc., USA).

## 3. Results

### 3.1. Isolated Fungi

Among the 47 fungal isolates from 23 *Dendrobium* orchid samples (13 from leaves, 22 from roots, 12 from stems), a total of 23 species (including 3 speculated novel *Xylaria* species and 2 undetermined species—*Induratia* sp. and *Xylariaceae* sp.) were identified based on phylogenetic analysis and morphology (Table 3). *Xylaria* species account for 51% of the isolated fungi. Nine new species from *Annulohypoxylon*, *Hypoxylon*, *Nemania*, and *Xylaria* were introduced. The newly introduced species are listed in Table 4.

### 3.2. Phylogenetic Analysis

The concatenated dataset of Apiosporaceae generated from the four loci comprised 129 taxa with 2044 characters including gaps (666 bp of ITS, 796 bp of LSU, 366 bp of TEF-1α, 216 bp of TUB2). The RAxML analysis for Apiosporaceae resulted in a final ML optimization likelihood value of −18,368.610931. The Bayesian tree converged at 1,045,000th generations with an average standard deviation of split frequencies of 0.009928. Another consensus phylogenetic tree was generated from the four loci concatenated datasets of other xylarialean strains with 404 taxa and 4107 characters including gaps (847 bp of ITS, 1432 bp of LSU, 1298 bp of RPB2, 530 bp of TUB2). The RAxML analysis resulted in a final ML optimization likelihood value of −188,105.524536. The Bayesian tree converged at 287,695,000th generations, with the average standard deviation of split frequencies 0.015.

The phylogram of Apiosporaceae presented three major monophyletic clades—*Apiospora*, *Arthrinium*, and *Nigrospora* (Figure 1). *Pseudomassaria* species were used as an outgroup with 94% ML/1.0BPP support. In *Apiospora*, *Apiospora dendrobii* (MFLUCC 14-0152) formed a sister clade to *Apiospora xenocordella* (ex-type CBS 478.86 and CBS 595.66) with 85% ML/1.0BPP support. In *Nigrospora*, *Nigrospora chinensis* comprising four strains—CGMCC 3.18127 (ex-type), LC4593, MFLUCC 14-0109, and MFLUCC 18-1215 closed to *Nigrospora camelliae-sinensis* LC4460 with 100% ML/1.0BPP support. *Nigrospora sphaerica* represented by three strains LC13523 (ex-type), LC7259, and GZAC O37S13 formed an independent clade.

Another phylogenetic tree is mainly composed of the Induratiaceae, Hypoxylaceae, and Xylariaceae species (Figure 2). A total of 43 strains were identified to 20 species distributed in five genera—*Annulohypoxylon* (1 species), *Hypoxylon* (4 species), *Induratia* (1 unresolved species), *Nemania* (3 species), *Xylaria* (10 species), and one unresolved species in genus incertae sedis of Hypoxylaceae.

Induratiaceae was a monophyletic group close to the genera *Anthostomella*, *Anthostomelloides*, and *Clypeosphaeria*. *Induratia* sp. (MFLUCC 15-1218) formed a single clade with low support. The Hypoxylaceae lineage presented a monophyletic clade comprising ten genera. *Annulohypoxylon*, *Daldinia*, and *Hypoxylon* were polyphyletic groups incorporating thirteen studied strains. *Annulohypoxylon moniliformis* (ex-type: MFLUCC 18-1214) and *An. annulatum* (ex-type: CBS 140775) clustered together with 99% ML/1.0BPP support. In the *Hypoxylon* part, *H. endophyticum* (ex-type MFLUCC 18-1206) encompassed six strains developed an independent clade adjacent to *H. investiens* with 100% ML/1.0BPP support. *H. officinalis* (ex-type MFLUCC 14-0075) was represented by three isolates separated from other species with 100% ML/1.0BPP support. *H. investiens* contained four strains with low support among each other. *H. pulicicidum* (ex-type MUCL 49879) consisting of four isolates formed a sister clade to *H. hinnuleum* (CBS 286.62). *Hypoxylaceae* sp. (MFLUCC 14-0141) was basal to mixed genera including *Anthocanalis*, *Daldinia* II and III, *Rhopalostroma*, and *Thamnomyces* with 74% ML/0.97BPP support.

In the Xylariaceae phylogram, according to Hsieh et al. (2010), four separated lineages representing the *Xylaria* ‘PO’ clade, *Nemania* and *Rosellinia* ‘NR’ clade, *Xylaria* ‘HY’ clade, and the genus *Pseudoxylaria* ‘TE’ clade naturally developed. *Xylaria hypoxylon* is the core species in the ‘HY’ clade. The ‘NR’ clade consists mainly of *Nemania* and *Rosellinia* species. ‘PO’ clade formed by *X. polymorpha* A total of 29 isolates distributed in *Nemania* and *Xylaria*. In the *Nemania* + *Rosellinia* ‘NR’ clade, *Nemania bipapillata* encompassing three isolates (90080610 HAST, MFLUCC 14-0105, and MFLUCC 14-0138) and *Nemania dendrobii* (ex-type MFLUCC 18-1213) clustered together with 100% ML/1.0BPP support, which formed a sister clade to *Nemania diffusa*. In the *X. cubensis* aggregate I of *Xylaria* ‘PO’ clade, *Xylaria berteri* including 11 strains separated from *Xylaria crozonensis* (398HAST) sustained by 99% ML/1.0BPP. *Xylaria* sp.1 and *X. cubensis* (GENT 159, 477 HAST, and JDR 860) grouped and developed an independent clade with low support. *Xylaria laevis* containing four strains was at the root of this aggregate. *Xylaria* sp. 2 (ex-type MFLUCC 21-0014) and *X. phyllocharis* clustered with 75% ML/0.97 BPP support. In the *X. corniformis* aggregate II, *Xylaria feejeensis* (GZAC O30S21) and *X. frustulosa* (92092010 HAST) gathered sustained by 77% ML/0.99 BPP. In the *X. corniformis* aggregate III, *X. curta* containing three isolates (GAZC O36L23, 494HAST, and 92092022 HAST) well separated from *X. karyophthora* (DRH059) with 100% ML/1.0BPP. In the *Xylaria* ‘HY’ clade, *X. arbuscula*, *X. apiculate*, and *X. venosula* formed a monophyletic clade close to *X. arbuscula* var. *plenofissura* (93082814 HAST) with 100% ML/1.0BPP support in the *X. arbuscula* aggregate. *Xylaria* sp. 3 (MFLUCC 21-0059) was a single branch sister to *X. bambusicola* sustained by 93% ML/0.97 BPP. Six strains pertaining to *Xylaria grammica* constituted an independent clade adjacent to *X. liquidambaris* (93090701 HAST) with low support. *Xylaria hongkongensis* (ex-type GDGM 40058) and *Xylaria papulis* 5118 adjoined to *X. mali* (CBS 385.35) with 100% ML/1.0BPP support.

### 3.3. Taxonomy

Xylariales [63].

Xylariales are commonly found as fungal endophytes in herbaceous plants, saprobes, or pathogens on rotting wood with a preference for humid, shady habitats and have a worldwide distribution [23,64]. Hyde et al. (2020) accepted 15 families in this order. Stroma is an important characteristic for distinguishing most of the genera in Xylariales. However, some genera are astromatic, and there are asexual Xylariales [19]. Many novel structures of interesting secondary metabolites have been discovered in species of this order, which indicates a great potential for investigating bioactive compounds [33].

Apiosporaceae K.D. Hyde, J. Fröhl., Joanne E. Taylor & M.E. Barr [65]

Apiosporaceae was introduced by Hyde et al. (1998) to accommodate *Apiospora* and Appendicospora according to the exclusive sexual morphology and special asexual morphs, such as basauxic conidiophores with terminal and intercalary polyblastic conidiogenous cells and unicellular conidia with germ slits [26]. Crous & Groenewald (2013) analyzed nrLSU DNA and confirmed Apiosporaceae taxonomic placement in Xylariales [20,65]. They are common fungal endophytes of various plants, and they can be saprobic and pathogenic for mostly monocotyledons and grasses [20]. Presently, six genera viz. *Appendicospora*, *Arthrinium*, *Dictyoarthrinium*, *Endocalyx*, *Scyphospora*, and *Spegazzinia* are accepted [20,34].

*Apiospora* Sacc. [66]

*Apiospora* Sacc. was proposed in 1875 and synonymized with *Arthrium* Kunze based on their basauxic conidiogenesis, large upper cell and small basal cell (sexual morph), LSU based (mainly) phylogenetic analysis following the one fungus-one name policy [65,67,68,69,70,71]. However, *Nigrospora* phylogenetically split the original *Arthrinium* species into two parts, which resulted in forming three monophyletic groups mainly due to their different sequences of ITS, LSU, TEF-1α, and TUB2 exons [72]. *Apiospora* was proposed to accommodate the species containing conidia that generally more or less rounded in face view and lenticular side view [72]. *Apiospora montagnei* (=*Sphaeria apiospora*) was the type species, however, the ex-type of strain and its sequences were absent [71,72]. This group of fungi are endophytes, pathogens, and saprobes associated with multiple plant host families worldwide [67,70,72]. *Apiospora phaeospermum* (published as *Arthrinium phaeospermum*) was reported as a human pathogen [73].

*Apiospora dendrobii* XY Ma & JC Kang, sp. nov. Figure 3.

Index Fungorum number: IF551811; Facesoffungi number: FoF 10263.

Etymology—Name after its host species-genus name.

Culture characteristics—Colonies on PDA superficial, white to light yellow hypha, circular, lightly undulate, radiant from the middle; reverse yellow. Vegetative hyphae septate, branched, hyaline, and thick-walled. Growth rate: 5 mm/day.

On 4% OA, vegetative hyphae hyaline was 1–4 μm diam., septate, smooth, branched, and thick-walled. Conidiophores were reduced to conidiogenous cells, hyaline to brown, thick-walled. Conidiogenous cells were erected from mature mycelia, aggregated, hyaline to brown, and straight to curved. Conidiomata 20 μm diam., superficial or immersed in aerial mycelia, globose to irregular, dark brown to black, aggregated. Conidia 7–11.5 × 9–12.5 μm ($\bar{x}$ = 9.5 × 10.3 μm, *n* = 50), hyaline to light brown, solitary, in surface view, ellipsoid to globose or sub-globose, lenticular from side view.

Material examined—THAILAND, Chiang Rai Province, Mae Fah Luang District, outside Temple of Doi Tung Pagoda, in the roots of *Dendrobium harveyanum*, 19 December 2013, S. Nontachaiyapoom, N. Aewsakul and X.Y. Ma. MFLU 21-0153 (Holotype); MFLUCC 14-0152 (ex-holotype strain).

Notes—The conidiogenous cell of *Apiospora dendrobii* is straight, which differs from that of *A. xenocordella* (verruculose, globose to clavate to doliiform). *Apiospora dendrobii* forms an individual branch clustered with *A. xenocordella* sustained by 100% ML/1.0BPP. The ex-type MFLUCC 14-0152 differs from the ex-type of CBS 478.86 of *A. xenocordella* by 1.95% (13/666 bp) of ITS and 3.24% (7/216 bp) of TUB2 sequences. The ITS blast search indicates that the most similar species is *Sordariomycetes* sp. (Sequence similarity 99%). Both LSU and TUB2 blast searches indicated that MFLUCC 14-0152 was the most similar species to *A. xenocordella* with sequence similarity of 99% and 94%, respectively. Although the morphological characteristics were not sufficient to designate the novel species, the base pair differences support it as a different species from *Apiospora xenocordella*.

*Nigrospora* Zimm. [74] 

*Nigrospora* are ubiquitous endophytes and saprobes, and some species are human and plant pathogens, with global distribution [55]. *Nigrospora* species are characterized by spherical to subspherical conidiogenous cells, black and globose to subglobose conidia [55,75]. Numerous secondary metabolites isolated from this genus have promising bioactive potentials, such as plant growth-inhibiting nigrosporolide and anti-mosquitoes phomalactone [76,77].

*Nigrospora chinensis* Mei Wang & L. Cai, Persoonia 39, 2017: 118–142 Figure 4

Culture characteristics—colonies on PDA were superficial, white, and cottony on the entire edge. Reverse white. Growth rate: 5 mm/day. Colonies on MEA white to brown, leathery, brown to black, conidiomatal masses scattered, reverse deep brown. Growth rate: 6 mm/day.

On 2% MEA, vegetative hyphae hyaline, 1.2–2.4 μm diam., septate, smooth, branched, and thick-walled. Conidiomata jelly-like, brown to black, aggregated, irregular. Conidia solitary, globose to subglobose, hyaline to light brown. 

Notes—*Nigrospora chinensis* (MFLUCC 14-0109 and MFLUCC 18-1215) was isolated from leaves of *Dendrobium cariniferum* collected from northern Thailand and southwestern China. The two strains clustered with ex-type of *Nigrospora chinensis* (CGMCC 3.18127) and another strain, LC4593, that were from *Camellia sinensis* in Guang Dong, China with 100% ML/1.0BPP support. Compared with the ex-type CGMCC 3.18127, MFLUCC 14-0109 and MFLUCC 18-1215 have very similar sequences and asexual morphological characteristics. The blast search also indicates that their most similar species is *Nigrospora chinensis* (Sequence similarity 99%).

*Nigrospora sphaerica* (Sacc.) [78]

Culture characteristics—colonies on PDA were superficial, white to light brown, velvety, zonate with one concentric circle, entire edge, reverse light brown. Vegetative hyphae septate, branched, hyaline, and thick-walled. Growth rate: 6.4 mm/day.

Notes—*Nigrospora sphaerica* (GZAC O37S13) was isolated from *Dendrobium hercoglossum* from Xingyi, Guizhou in southwestern China. It failed to sporulate on PDA after one month. From the phylogeny, the isolate GZAC O37S13 and two *Nigrospora sphaerica* strains, including the ex-type, were isolated from Musa paradisiaca in Hai Nan, China. Compared with the ex-type of LC 13523 of *Nigrospora sphaerica*, the sequences of GZAC O37S13 are similar to that of LC 13523. The robust support value 100% ML/1.0BPP exhibited their intimate relationship. 

Hypoxylaceae DC. [79]

Hypoxylaceae was resurrected by Wendt et al. (2018) following a multi-locus phylogeny to accommodate nodulisporium-like asexual genera, which was accepted by Daranagama et al. (2018) and Hyde et al. (2020). Most hypoxylaceous species are endophytes and saprobes on herbaceous and woody plants, while some species are linked with insect vectors [19,31,37,76].

*Annulohypoxylon* Y.M. Ju, J.D. Rogers & H.M. Hsieh [19]

*Annulohypoxylon* was introduced by Hsieh et al. (2005) to accommodate the section Annulata of *Hypoxylon* [80]. Most *Annulohypoxylon* species are saprobic on decorticated and corticated wood, and some species are endophytes of herbaceous plants [23,37]. Wendt et al. (2018) amended the morphological characteristics of *Annulohypoxylon* and established the genus *Jackrogersella* as a separate group from *Annulohypoxylon* based on chemotaxonomy, morphology, and multigene genealogy. The *Annulohypoxylon* became polyphyletic since *Rostrohypoxylon* was introduced [80,81,82]. *Annulohypoxylon* sensu stricto is mostly characterized by ostioles encircled by an annulated disc, binaphthalenes as major stromal metabolites, and a lack of azaphilones [19,20]. As endophyte, *Annulohypoxylon* spp. Have been isolated from *Dendrobium chrysotoxum*, *D. crystallinum*, *D. falconer*, and *Stanhopea trigrina* [37].

*Annulohypoxylon moniliformis* XY Ma & JC Kang, sp. nov. Figure 5.

Index Fungorum number: IF551812; Facesoffungi number: FoF 10264.

Etymology—Named after its host species.

Culture characteristics—colony on PDA was superficial, white, cottony, undulate edge, reverse white to light brown. Growth rate: 5.7 mm/day.

On WA vegetative hyphae 1.7–3.5 μm diam., hyaline to brown, septate, branched, smooth to rough, from young to old, thick-walled. Ascomata 200 × 600 μm was wide, black, on rubber band of slide culture, superficial, cylindrical to irregular. Ostioles 69–142 × 6.3–8.4 μm, hyaline to light brown, lower than the stromal surface. Asci was cylindrical and thick-walled. Ascospores 7–8.5 × 4–6 μm ($\bar{x}$ = 7.5 × 5 μm, *n* = 5), hyaline, ellipsoidal to fusiform with narrowly rounded ends, and thick-walled.

Material examined—CHINA, Guizhou Province, Gui Yang City, Luodian County, in the roots of Dendrobium aphyllum, 27 November 2015, J.C. Kang. GZAC O35L22 (Holotype); MFLUCC 18-1214 (ex-holotype strain).

Notes—*Annulohypoxylon moniliformis* and *An. annulatum* clustered with 98% ML/1.0BPP support. It is difficult to judge their morphology because our characteristics are insufficient and unclear, although they differ by color and size of ascospores (hyaline 7–8.5 × 4–6 μm in 1 month (*An. moniformis*) vs. brown to dark brown, unicellular, 7.5–11 × 3.6–6 μm in 2–3 weeks (*An. annulatum*)) on different media (WA of slide culture vs. OA). The blast search of all sequences indicated that *An. annulatum* was the most similar species to *An. moniliformis* (Sequence similarity 99%). However, a pairwise nucleotide comparison between the MFLUCC 18-1214 and *An. annulatum* (ex-type CBS 140775) revealed 1.53% (13/847 bp) of ITS and 4.51% (42/932 bp) of TUB2 sequence differences. Therefore, based on the principal that more than 1.5% nucleotide differences in the ITS gene perhaps indicate a new species [83], we introduce MFLUCC 18-1214 as a new species based on the base pair differences.

*Hypoxylon* Bull. [84]

*Hypoxylon* is the type of genus of Hypoxylaceae, which mainly inhabit dead wood as saprobes, and some occur as endophytes of a wide range of hosts or facultative parasites on diseased hosts. They have *Nodulisporium*-like asexual morphs [20,80]. Stomatal (including pigment) and ascospore morphology were often used for morphological delimitation of the *Hypoxylon* species [20]. Multigene genealogy revealed that they are a polyphyletic group [19,67]. Many bioactive secondary metabolites have been identified from *Hypoxylon* species, especially from endophytic strains [33,85,86].

*Hypoxylon endophyticum* XY Ma & JC Kang, sp. nov. Figure 6.

Index Fungorum number: IF551815; Facesoffungi number: FoF 10265.

Etymology—Name after its life strategy as a fungal endophyte.

Culture characteristics—Colony on PDA superficial, greyish to white, velvety, loose, and on entire edge; reverse brown to black. Growth rate: 5.6 mm/day.

On PDA vegetative hyphae 1.6–4.5 μm, hyaline to brown, smooth to rough, and branched. On WA (slide culture), conidiomata superficial, black, and irregular. Conidiophores macronematous, cylindrical, hyaline to brown, septate, branched, and finely roughed. Conidia 3–5 × 1.5–3.5 μm ($\bar{x}$ = 4 × 2.5 μm, *n* = 30), globose to subglobose, hyaline to brown. 

On WA (slide culture), vegetative hyphae 2–5 μm, hyaline to brown, smooth to rough, and branched. Conidiomata 200–500 μm wide, light brown, and irregular. Conidiophores mononematous or macronematous, Periconiella-like, cylindrical, hyaline to brown, integrated, septate, branched, and finely rough. Conidia 3.5–6.5 × 2–5.5 μm ($\bar{x}$ = 5.5 × 3 μm, *n* = 20), hyaline, globose to ellipsoid, sympodially, and smooth. 

Material examined—CHINA, Guizhou Province, Luodian County, Orchid nursery, in the roots of *Dendrobium loddigesii* and stems of *D. huoshanense*, 4 April 2016, B.W. Chen, living cultures, MFLUCC 18-1209, MFLUCC 18-1210, MFLUCC 18-1206; China, Guizhou Province, Xingyi City, Orchid nursery, in the roots of *Dendrobium aphyllum* and *Dendrobium* sp., leave of *D. hercoglossi* and *D. chrysotoxum*, 27 November 2015, J.C. Kang, MFLUCC 18-1207, MFLUCC 18-1208, MFLUCC 18-1211. MFLU 21-0154 (Holotype); MFLUCC 18-1206 (ex-holotype strain). 

Notes—*Hypoxylon endophyticum* formed an independent clade close to *H. investiens*. The sequence variation among the six strains is less than 1% except for the MFLUCC 18-1207 that may be caused by the crossing contamination or intraspecific variation. *Hypoxylon endophyticum* failed to be discriminated with *H. investiens* although the size of conidia in this study are larger than that of *Hypoxylon investiens* in Ju & Rogers (1996) considering different media (3.5–6.5 × 2–5.5 μm on WA in 1 month vs. 2.5–3.5 × 2.2–3.5 μm on OA in 2 weeks) [78]. The distinction based on base pairs between *Hypoxylon endophyticum* and *H. investiens* have been listed in Table 4. The large difference exists in the sequences of LSU and RPB2. Blast searches for the six isolates showed that *Hypoxylon investiens* is the most similar species with sequence similarity from 96–97%.

*Hypoxylon officinalis* XY Ma & JC Kang, sp. nov. Figure 7.

Index Fungorum number: IF551818; Facesoffungi number: FoF 10268.

Etymology—Named after it host epithet (*Dendrobium officinale*).

Culture characteristics—Colony on PDA superficial was white, buff, or amber, velvety or cottony, sometimes radial, with or without a concentric ring, denticulate with an entire or loose edge, and reverse unevenly white to brown. Growth rate: 6.7 mm/day.

On PDA, vegetative hyphae 1.7–3.5 μm diam., hyaline to brown, smooth to finely roughed, branched, septate, and swollen at mycelia nodes. On MEA, conidiomata 1 cm high., aggregates white to brown, and was cylindrical. Conidiophores mononematous or macronematous was cylindrical, hyaline to brown, integrated, septate, and finely roughed. Conidia 5.5–7 × 2.5–3.5 μm ($\bar{x}$ = 6.1 × 3 μm, *n* = 3) was hyaline to brown, ellipsoidal to reniform, and smooth-walled.

Material examined–THAILAND, Chiang Rai Province, Mae Fah Luang District, outside Temple of Doi Tung Pagoda, in the roots of unidentified *Dendrobium* sp., 19 December 2013, S. Nontachaiyapoom, N. Aewsakul and X.Y. Ma, MFLUCC 14-0075, MFLUCC 14-0078; China, Guizhou Province, Gui Yang City, Luodian County, in the roots of *Dendrobium aphyllum*, 27 November 2015, J.C. Kang, MFLUCC 21-0060. MFLU 21-0152 (Holotype); MFLUCC 14-0075 (ex-holotype strain). 

Notes—*Hypoxylon officinalis*, represented by MFLUCC 14-0075, MFLUCC 14-0078, and MFLUCC 21-0060, formed an independent clade adjacent to *Hypoxylon lateripigmentum* (MUCL 53304). The three strains were all from *Dendrobium* roots. Compared with the ex-type MUCL 53304, *Hypoxolon officinalis* was not observed having the *periconiella*-like conidiogenous structure, and the shape of conidia is a bit more changeable. Their sequence base pair discrepancy is listed in Table 4.

*Hypoxylon investiens* (Schwein.) M.A. Curtis [87] Figure 8.

Culture characteristics—Colony on PDA superficial, white to brown, velvety, mycelia gather unevenly, with an entire edge, reverse brown to black, and radial. Growth rate: 7 mm/day.

On MEA and WA (slide culture), vegetative hyphae 1.2–3.2 μm diam., hyaline to brown, smooth to rough, and branched. On MEA, conidiomata was superficial or immersed, scattered, black, and globose to irregular. Ascospores was brown to dark brown, sub-globose, ellipsoid and clavate, nearly equilateral, with one or two rounded ends, faint, and straight germ slit shorter than spore-length. Conidia on WA 2–4.5 × 1.5–3 μm ($\bar{x}$ = 3.5 × 2.2 μm, *n* = 20), hyaline, globose to ellipsoidal, smooth-walled, thick-walled, with or without guttules. 

Material examined—THAILAND, Chiang Rai Province, Mae Fah Luang District, outside Temple of Doi Tung Pagoda, in the stem of *Dendrobium moschatum*, 11 May 2015, S. Nontachaiyapoom, B. Mala and X.Y. Ma

Notes—The strain MFLUCC 15-1155 was isolated from the stems of *Dendrobium moschatum* collected from northern Thailand. It clustered with three *Hypoxylon investiens* strains sustained by low support. Type specimens with sequence data of *Hypoxylon* investiens is absent. Contrasted to the morphological descriptions of *Hypoxylon investiens* in Ju and Rogers (1996), MFLUCC 15-1155 is the same as *Hypoxylon investiens* although periconiella-like conidiogenous structure was not observed on MFLUCC 15-1155, and the latter has smaller conidia perhaps resulting from different media. However, the ITS base pair discrepancy among the four *Hypoxylon investiens* strains—YMJ 89062905, CBS 118185, CBS 118183, and MFLUCC 15-1155 can be up to 3.5%. We speculate that a complex exists which needs further verification and comparison with confirmed type species and molecular data. The blast search shows that *Hypoxylon investiens* has the highest 96% sequence similarity with strain MFLUCC 15-1155. 

*Hypoxylon pulicicidum* J. Fournier, Polishook & Bills [88] Figure 9C.

Culture characteristics—Colony on PDA was superficial, white to amber, velvety, entire edge, and reverse brown to black. Growth rate: 6 mm/day.

Notes–The strain GZAC O37L14 was isolated from leaves of *Dendrobium hercoglossum* collected from southern China. Compared with the ex-type MUCL 49879 of *Hypoxylon pulicicidum*, it has little sequence difference (1.3% (8/589 bp) of ITS and 0.38% (1/265 bp) of TUB2).

*Hypoxylaceae* sp. (Unresolved species) Figure 10.

Culture characteristics—Colonies on PDA were superficial white, irregular, with lobbed concentric ring and undulate margin, flossy, velvety, reverse white, with the ununiform concentric ring. Growth rate: 5 mm/day.

On 4% OA, Vegetative hyphae was 2.14–3.49 µm diam., hyaline, smooth, septate, branched, and thick-walled. Conidiophore-like structures, branched, interminate, and crooked. β conidia-like structures were straight or curved.

Material examined: Thailand, Chiang Rai Province, Mae Fah Luang District, outside Temple of Doi Tung Pagoda, in roots and stems of *Dendrobium* spp., 19 December 2013, S. Nontachaiyapoom, N. Aewsakul and X.Y. Ma, MFLUCC 14-0141. 

Notes: *Hypoxylaceae* sp. (MFLUCC 14-0141) is a solo branch taxon basal to *Daldinia* II & III and another three genera with 74% ML/0.97BPP support in Hypoxylaceae. Sporulation failed with CMA, MEA, OA, PDA, SNA, and WA. Compared with *Thamnomyces dendroidea* (CBS 123578), it has 8.03% (68/847 bp) of ITS, 4.47% (64/1432 bp) of LSU, and 19.4% (103/530 bp) of TUB2 different sequences. The blast search for three gene sequences showed different results—*Xylaria enteroleuca* (CBS 128357) with 100% ITS sequence similarity, *Rhopalostroma indicum* (CBS 113035) with 99.6% LSU sequence similarity) and *Daldinia brachysperma* (BCC 33676) with 90.4% TUB2 sequence similarity. Excluding contamination, we speculate that the isolate MFLUCC 14-0141 is a cryptic taxon needing further research.

Induratiaceae Samarak., Thongbai, K.D. Hyde & M. Stadler [89].

Induratiaceae was introduced by Smarakoon et al. (2020) to accommodate *Emarcea* and *Induratia* (=*Muscodor*) with apiospores and independent phylogenetic clade trees. This group is usually saprobic on dead wood and leaves and endophytic on leaves, stems, and bark [89]. The hyphae of Induratiaceae are rope-like with cauliflower-like hyphal bodies [89].

*Induratia* Samuels, E. Müll. & Petrini [89]

= *Muscodor* Worapong, Strobel & W.M. Hess, Mycotaxon 79:71 (2001).

*Moscodor* is an endophyte genus without morphological record and established only based on molecular data, which can produce volatile antibiotics and be distinguished from each other by their chemical profiles [23]. However, the genus was not accepted by Stadler et al. (2013) and Wendt et al. (2018). It was listed as Xylariales genera incertae sedis by Maharachchikumbura et al. (2016) and Daranagama et al. 2018. *Induratia* and *Muscodor* were thought to be the same genus corresponding to the sexual and asexual states in phylogenetic studies [88]. *Induratia* was adopted as the genus name according to the One-Fungus-One-Name proposal [89]. This group is saprobic on dead wood and endophytic on bark, leaves, roots, and stems [89]. The asexual morph is characterized by terminal conidiogenous cells bearing inconspicuous denticles, conidia narrowly ellipsoidal to subglobose, hyaline, smooth-walled, with a flat, wide, basal scar [89].

Induratia sp. (Unresolved species) Figure 9B.

Culture characteristics—Colony on PDA was white, cottony, undulate edge, and reverse light brown. Growth rate: 3.6 mm/day.

Notes—*Induratia* rarely produces morphological structures on artificial media. *Induratia* sp. was isolated as a fungal endophyte from the root of *Dendrobium nobile* [90]. The endophytic strain MFLUCC 15-1218 was isolated from the roots of *Dendrobium* sp. and collected from a steep forest in northern Thailand. It clustered with *Induratia brasiliensis* and *I. ziziphi*. However, it is hard to discern its taxonomic placement based on the ITS sequences with less than 1% base pairs difference. 

Xylariaceae Tul. & C. Tul. [19]

Xylariaceae is one of the largest families of Xylariales and can be a saprobe, pathogen, or endophyte on a wide range of hosts, substrates, or associated insect vectors [20,23]. Many xylariaceous species have been founded as endophytes and involved in the study for natural products [11,18,23,33]. The stromata and ascomata of Xylariaceae are variable in size [19,20]. Their sexual morph is hyphomycetous, which are mostly geniculosporium-like [20,23]. Hyde et al. (2020) accepted 32 genera in Xylariaceae.

*Nemania* Gray [91]

*Nemania* is a large genus of Xylariaceae and characterized by stromata without bark rupturing appearance, a lack KOH-extractable pigments, and finely papillate ostioles [20]. Maharachchikumbura et al. (2016) and Réblová et al. (2016) proposed the use of *Nemania* over *Geniculosporium* following Stadler et al. (2013). They are mostly found as saprobe and endophytic on woody and herbaceous plants [11,23]

*Nemania dendrobii* XY Ma & JC Kang, sp. nov. Figure 11.

Index Fungorum number: IF551819; Facesoffungi number: FoF 10269.

Etymology—Name after its host genus.

Culture characteristics—colonies on PDA were superficial, white, felt, undulate edge, with yellow centers, and reverse brown. Growth rate: 6 mm/day. Colonies on WA were velvety, with concentric grey rings, a little brown spot scattered, undulate edge, and reverse brown to white from the inner part to edge. Growth rate: 4.5 mm/day.

On 4% OA, vegetative hyphae, 0.7–1.2 µm diam., hyaline, straight, branched, aseptate, finely-roughed, with rich lipid droplets. Conidiophores rising from mycelia, mononematous, crooked, and finely roughed. Conidia ellipsoid, hyaline, and guttules.

Material examined—CHINA, Guizhou Province, Xingyi City, Orchid nursery, in the roots and stems of *Dendrobium* spp., 4 October 2016, B.W. Chen, MFLUCC 18-1212, MFLUCC 18-1213. GZAC O49S1A (Holotype); MFLUCC 18-1213 (ex-holotype strain).

Notes—The two isolates MFLUCC 18-1212 and MFLUCC 18-1213 have identical sequences and form a distinct clade close to *Nemania bipapillata* 90080610 (HAST) with 100% ML/1.0BPP support. However, the type specimens with molecular data are absent. The morphological characteristic is scanty for identification and comparison. The big difference can only be observed in sequence differences (Table 4). The blast search for each gene of MFLUCC 18-1212 and MFLUCC 18-1213 revealed *Nemania bipapillata* is the most corresponding species with similarities of 93%–95%.

*Nemania bipapillata* (Berk. & M.A. Curtis) Pouzar, Ceská [92] Figure 12.

Culture characteristics: colonies on PDA were superficial, white, leathery, lobbed and stacked, undulate edge, and reverse white. Growth rate: 4.5 mm/day.

On PDA, vegetative hyphae 1.7–3.3 μm diam., hyaline to brown, smooth, branched, and thick-walled. Conidiomata 500 × 1000 μm were scattered, brown to black, and clavate to irregular. Conidiophores were cylindrical, sometimes swollen in the upper part, integrate, interminate, macronematous erect, septate, branched or unbranched, smooth, and thick-walled. Conidiogenous cell holoblastic, brown. Chlamydospores were ellipsoidal, hyaline, and thick-walled. Conidia 5.5–9 × 3.5–6 μm ($\bar{x}$ = 6.7 × 4.6 μm, *n* = 8) was brown, globose to ellipsoidal, truncate at the end, and thick-walled.

Notes—*Nemania bipapillata* comprising three isolates MFLUCC 14-0105, MFLUCC 14-0138, 90080610 (HAST) were well separated with *N. dendrobii*. The two strains, MFLUCC 14-0105 and MFLUCC 14-0138, were isolated from stems of *Dendrobium cariniferum* and roots of *Dendrobium* sp. collected from northern Thailand. The asexual morphological descriptions and type molecular data are absent. The blast search showed that *Nemania bipapillata* matched each gene with a high sequence similarity of 97–100%.

*Nemania diffusa* (Sowerby) Gray, [91] Figure 13.

Culture characteristics—colonies on PDA were superficial, white, cottony, radial, entire edge, exudate colorless, and reverse white. Growth rate: 7 mm/day. 

Description—Vegetative hyphae 1–3 μm on 2% MEA were septate, hyaline to brown, smooth, branched, and thick-walled. Conidiophore-like structures were cylindrical, hyaline to brown, smooth, and thick-walled. Conidia-like structures were ellipsoidal, hyaline to brown, and thick-walled. 

Note—The strain MFLUCC 14-0139 was isolated from the root of *Dendrobium* sp. collected from Thailand. *Nemania diffusa* represented by three strains 91020401 (HAST), JZB3370003, and MFLUCC 14-0139 formed a single clade close to *Hypoxylon argillaceum* (CBS 527.63) with robust support (100% ML/1.0BPP). *Hypoxylon argillaceum* (CBS 527.63) is the only hypoxylaceous species in the ‘*Nemania* + *Rosellinia*’ clade. The same situation also occurred in U’ren et al. 2016 and re-evaluation was recommended. We speculate that CBS 527.63 should be ‘*Nemania argillaceum.’* Although MFLUCC 14-0139 holds 3.8% of the different TUB2 sequences compared with 91020401 (HAST), considering their identical ITS sequences and scanty morphological information, incorporating the blast search for each gene, we identified it as *Nemania diffusa*. 

*Xylaria* Hill ex Schrank [93]

*Xylaria* is a genus type of Xylariaceae and is characterized by large stromata, long asci with stipes, dark ascospores, and geniculosporium-like asexual morph [20,85,90]. The highly diversified nature of the genus may be the result of highly convergent evolution within the genus [19,20]. Most *Xylaria* are saprobic on deciduous dead wood and endophytes of numerous plants, especially pantropical areas, some associated with termites [23].

*Xylaria* sp.1 Figure 14.

Index Fungorum number: IF 551823; Facesoffungi number: FoF 10273.

Culture characteristics—colonies on PDA were superficial, white, cottony, undulate edge, zonate with one concentric ring, scattered aggregated white spots, and reverse white to buff in the concentric ring. Growth rate: 4.7 mm/day.

On 4% OA, vegetative hyphae 1.5–3.3 μm were hyaline, smooth, septate, branched, and thick-walled. Conidiomata were 750 μm wide, solitary or aggregate, white-orange to orange, and irregular. Conidiophore were hyaline, palisades, mononematous or macronematous, with abundant lipid droplets, branched near the base, smooth, and thick-walled. Conidiogenus cells were cylindrical, terminate, Conidia 4.5–9 × 2.5–4.5 μm ($\bar{x}$ = 6.2 × 3.1 μm, *n* = 24), hyaline, globose to ellipsoid, or reniform or cone-like, guttulate, and smooth.

Material examined—THAILAND, Chiang Rai Province, Mae Fah Luang District, outside Temple of Doi Tung Pagoda, in the roots of *Dendrobium* sp., 19 December 2013, S. Nontachaiyapoom, N. Aewsakul and X.Y. Ma, MFLUCC 14-0137. China, Guizhou Province, Luodian County, orchid nursery, in the leaves of *Dendrobium officinale*, 1 December 2014, J.C. Kang, GZAC O6LA2. 

Notes—*Xylaria* sp.1 (MFLUCC 14–0137 and GZAC O6LA2) is adjacent to the isolate GENT 159 of *X. cubensis*, which forms a sister group to another two *X. cubensis* isolates. No type of herbarium specimens for *X. cubensis* was assigned. Compared with the asexual morph of *Xylaria cubensis* recorded by Rodrigues et al. (1993), *Xylaria*. sp. 1 has much bigger conidia with no obvious denticulate secession scars on the conidiogenous cell and a flat basal abscission scar on conidia. In gene sequence comparison, as listed in Table 4, *Xylaria* sp.1 and *X. cubensis* mainly differed by the ITS and TUB2 gene sequences. Here we introduce both MFLUCC 14–0137 and GZAC O6LA2 as a new species separated with *Xylaria cubensis*. *X. cubensis* isolate GENT 159, and *Xylaria* sp.1 MFLUCC 14-0137 has only six distinguished base pairs in the TUB2 sequence. However, GENT 159 is more different from the other two *X. cubensis* isolates. The taxonomic placement of GENT 159 perhaps needs re-evaluation with other gene sequence data.

*Xylaria* sp. 2 Figure 15.

Index Fungorum number: IF551822; Facesoffungi number: FoF 10272.

Culture characteristics—colonies on PDA were superficial, white, velvety, aerial mycelia, on the entire edge, and reverse white. Vegetative hyphae were septate, branched, hyaline, and thick-walled. Growth rate: 6.5 mm/day.

On 4% OA, vegetative hyphae had a 1–3.5 μm diam., were hyaline, smooth, branched, septate, with swollen nodes and abundant lipid droplets, and thick-walled. Conidiophores were hyaline, mononematous, crooked, integrate, septate, smooth, and thick-walled. Conidia were hyaline, ellipsoidal to reniform, smooth-walled, and thick-walled.

Material examined—CHINA, Guizhou Province, Gui Yang City, Animal husbandry and veterinary institute, in the stems of *Dendrobium chrysanthum*, 11 April 2016, S.X. Zhou and X.Y. Ma. MFLUCC 21-0014.

Notes—The conidia and conidiophores of *Xylaria* sp.2 sporulated on 4% OA are similar to the asexual morph of *Xylaria* [45]. In view that only two conidia were observed in this culture, the conidia size was not measured because the sample was too small to be representative. *Xylaria* sp. 2 clusters with *X. phyllocharis*. However, both type specimens of *Xylaria phyllocharis* and its asexual morph are absent. The strong evidence for the unique taxonomic placement of *Xylaria* sp. 2 is from the phylogenetic analysis (Table 4). *Xylaria cubensis*, *X. digitata*, and *X. enteroleuca* are the best match in blast search for MFLUCC 21-0014 with sequence similarities from 85.5–98% (ITS could not match a known species).

*Xylaria* sp. 3. Figure 16.

Index Fungorum number: IF551821; Facesoffungi number: FoF 10271.

Culture characteristics—colonies on OA were superficial, white, velvety, mycelia aerial and radiate, forming condense white spots, entire edge; reverse light brown, brown-black masses, and spots. Growth rate: 3.5 mm/day.

On PDA and 4% OA, vegetative hyphae 1–2.2 μm diam., hyaline to brown, smooth to finely roughed, branched, septate, and thick-walled. On OA, vegetative hyphae 0.9–2.5 μm diam., hyaline, smooth, branched, septate, thick-walled. Conidiomata were superficial, globose to irregular, aggregates, and white to brown. Conidiophores were mononematous or macronematous, cylindrical, hyaline to brown, integrate, and smooth to finely roughed. Conidia 4.5–8 × 2–4 μm ($\bar{x}$ = 5.7 × 2.8 μm, *n* = 25) were hyaline, ellipsoidal to reniform, smooth-walled, and thick-walled.

Material examined—China, Guizhou Province, Luodian County, isolated from the roots of Dendrobium aphyllum, 27 November 2015, J.C. Kang. MFLUCC 21-0059. 

Notes—The blast search for each gene of the strain MFLUCC 21-0059 indicated that *Xylaria bambusicola* is the best match with sequence similarities from 89–98%. Despite few available asexual morph records about *X. bambusicola*, the observed conidial characteristics resemble the asexual descriptions for endophytic *Xylaria* [45]. Erroneous or low-quality sequencing was excluded by checking sequence chromatogram and consensus blast results for each gene. We speculated that MFLUCC 21-0059 represents a tentative new species.

*Xylaria berteri* (Mont.) Cooke ex J.D. Rogers & Y.M. Ju, N. [94] Figure 9H and Figure 17.

Culture characteristics—colonies on PDA were superficial, white, velvety or cottony, stacked, lobbed or entire or undulate edge, and reverse white to brown. Growth rate: 4 mm/day. 

On PDA, vegetative hyphae 2.7–4.8 μm, hyaline to brown, septate, smooth, branched, and thick-walled. Conidiomata were superficial, erect through the media, globose, with flattened apex, and white to light brown. Conidiophores had a 4.4 μm diam., were cylindrical to clavate, hyaline to brown, branched or unbranched, smooth, and sometimes laterally compressed into a tight layer. Conidia were hyaline to brown, ovoid to ellipsoid, with granular contents, smooth-walled, thick-walled, with a flattened basal scar.

On WA (1 m^2^ WA cube on slide culture), vegetative hyphae 0.8–2.5 μm, wrer hyaline to light brown, septate, and smooth. Conidiomata semi-immersed, erect from media, irregular, surrounded by white mycelia, with concave ostioles. Conidiophores were string-like and made from a bunch of consecutive fusiform cells, hyaline to light brown, branched, and smooth. Conidia 8–12 × 2–4 μm ($\bar{x}$ = 10 × 3 μm, *n* = 20) were hyaline, ellipsoidal to fusiform, produced at the tip of conidiophore or laterally, single or percurrent, with one round end and blunt base, smooth-walled, another end truncated with flat basal abscission scar, and thick-walled.

Notes—*Xylaria berteri* (strains MFLUCC 21-0061, MFLUCC 14-0095, MFLUCC 14-0102, MFLUCC 14-0110, MFLUCC 14-0117, MFLUCC 14-0126, MFLUCC 14-0143, MFLUCC 14-0150, MFLUCC 14-0158) were isolated from leaves, roots, and stems and of *Dendrobium cariniferum*, *D. harveyanum*, and *Dendrobium* spp. collected from northern Thailand. All strains have identical sequences. *Xylaria berteri* has no assigned type specimens and asexual morph records. The blast results of each region indicated that *X. berteri* is the best match species (Sequence similarity 99–100%). Based on these, we identified these strains as the species *X. berteri*.

*Xylaria curta* Fr. [95] Figure 9G.

Culture characteristics—colonies on PDA were superficial, white, cottony aerial hyphae and brown concentric rings in the center, leathery outside the center, radial with bunched mycelia, denticulate edge, and reverse light brown. Growth rate: 5 mm/day.

Notes—As a fungal endophyte, Xylaria curta was isolated from the roots of *Dendrobium aphyllum* and *D. chrysanthum* [37]. Although GZAC O36L23 and 92092022 (HAST) have 7.1% of distinct ITS base pairs with 1.3% and 1.1% of sequence differences in TUB2 and RPB2, respectively, we tend to identify GZAC O36L23 as *Xylaria curta* due to the phylogenetic placement and scanty morphology. The ITS blast results showed that *Xylaria* cf. *curta* (K.-L. Chen L148) was the best match species with 94.2% sequence similarity. *Xylaria* cf. *curta* (K.-L. Chen L148) represents an endophyte isolated from leaves of lotus [96]. However, the classification of *Xylaria* cf. *curta* is unclear. The TUB2 and RPB2 gene sequence blast search indicated that *X. curta* is the most similar species to GZAC O36L23 (Sequence similarity 93–99%). We speculate that GZAC O36L23 might be an intraspecific variant of Xylaria curta.

*Xylaria feejeensis* (Berk.) Fr. [95]. Figure 9E.

Culture characteristics—colonies on PDA were superficial, white, velvety, radial, zonate with several concentric circles, entire edge, and reverse unevenly white to light brown. Vegetative hyphae were septate, branched, hyaline, and thick-walled. Growth rate (on PDA): 6 mm/day.

Notes—*Xylaria feejeensis* is an endophyte isolated from the roots of *Dendrobium fimbriatum* and *D. crystallinum* [37]. In this study, *Xylaria feejeensis* (GZAC O30S21) was isolated from the stems of *Dendrobium aurantiacum* Rchb. f. var. *denneanum* sampled in Xingyi City, Guizhou, China. It has an identical ITS sequence as JDR 180. The blast results show that the most similar species is *X. feejeensis* (Sequence similarity 97–99%).

*Xylaria laevis* Lloyd [97] Figure 9F.

Culture characteristic—colonies on PDA were superficial, white, velvety, radial, zonate with several concentric circles, entire edge, and reverse white and brown around some marginal parts. Vegetative hyphae were septate, branched, hyaline, and thick-walled. Growth rate: 4.5 mm/day.

Notes—Two target strains, GZAC O33L12 and GZAC O6LA2, were isolated from the leaves of *Dendrobium aurantiacum* and *D. officinale* sampled in Xingyi City, Guizhou, China. The gene blast search showed that *Xylaria laevis* was their best match species. (Sequence similarity 95–98%). Despite gene sequence disparity between the two isolates, due to a lack in further evidence, both are identified as *Xylaria laevis*.

*Xylaria grammica* (Mont.) Mont. [98] Figure 18.

Culture characteristics—colonies on PDA wre superficial, white to light brown, zonate with concentric circle, with velvety to cottony, radial, entire or slightly undulate edge, and reverse white to dark brown. Growth rate: 6 mm/day.

Description—Vegetative hyphae were 2–3.6 μm, hyaline to brown, septate, smooth, branched, and thick-walled. Conidiophores-like structures were 3.2–5.1 μm wide, cylindrical, hyaline, branched, and septate. Conidiogenous cells were cylindrical and holoblastic. Conidia-like structures were 13.5–32.5 × 5–8 μm ($\bar{x}$ = 19.5 × 6.7 μm, *n* = 10), hyaline to brown, ellipsoid or pyriform to the gourd, septate, with one end rounded and the other truncate.

Notes—*Xylaria grammica* has been isolated as endophyte from the root of *Dendrobium aphyllum*, *D. nobile*, *D. chrysanthum*, *D. chrysotoxum*, *D. crystallinum*, and *D. fimbriatum* [37,84]. In this study, *Xylaria grammica* (MFLUCC 14-0093 and MFLUCC 14-0146) were isolated from leaves of *Dendrobium* sp. collected from northern Thailand. The mononematous conidiophores and conidia with abscission scar resembled the asexual morph of *Xylaria*, but the conidia are much larger with more variant shapes [45]. Although lacking type specimens, they have identical sequences as *Xylaria grammica* (Strains BCC 1002 and 5228) and all cluster with well-supported values (100% ML/1.0BPP). The blast results for the two strains showed that *Xylaria grammica* has the highest sequence similarity of 99–100%.

*Xylaria papulis* Lloyd [99] Figure 9F.

Culture characteristics—colonies on PDA were superficial, white, velvety in the center, leathery outside the center, radial with bunched mycelia, one concentric circle, denticulate edge, and reverse light brown. Vegetative hyphae were septate, branched, hyaline, and thick-walled. Growth rate: 4.2 mm/day.

Notes—*Xylaria populis* GZAC O32S24 was isolated from the stems of *Dendrobium chrysotoxum* collected from Luodian County, Guizhou Province, southwestern China. *Xylaria papulis* 5118 as an endophyte was isolated from the roots of *Dendrobium aphyllum*, *D. chrysotoxum* and *D. fimbriatum* [37]. In view that the isolate GZAC O32S24 is identified as *Xylaria papulis* with only ITS sequence in the polyphasic analysis, it may not be a stable result. The ex-type of *Xylaria hongkongensis*, GDGM 40058, has less than 0.5% ITS sequence discrepancy with *Xylaria papulis* 5118. Therefore, *Xylaria hongkongensis* GDGM 40058 perhaps need to be re-evaluated with further gene sequences.

*Xylaria venosula* Speg. [100]. Figure 19.

=*Xylaria arbuscula* CBS 126415, CBS 126416

=*Xylaria apiculata* EF6

Culture characteristics—colonies on PDA were superficial, white to light yellow, velvety, had an entirely or slightly undulated edge, and were reverse light brown to dark brown. Vegetative hyphae were septate, branched, hyaline, and thick-walled. Growth rate: 4.3 mm/day. Colonies on MEA were white, velvety, on the entire edge, with 1–2 cm high charcoal stroma, stroma cylindrical with a white top, and reverse white with black spots. Growth rate: 4.5 mm/day. Stroma formed after 2 months.

On 4% OA, vegetative hyphae were septate, branched, hyaline, and thick-walled. Conidiophore were hyaline, palisades, mononematous, smooth, and thick-walled. Conidia 3–5 × 2–4 μm ($\bar{x}$ = 5.8 × 3.3 μm, *n* = 10) were hyaline, globose to ellipsoid, guttules, smooth-walled, and thick-walled. The appressoria-like structure rising from mycelia was crooked, septate, and had crescent head.

Notes—*Xylaria venosula* were isolated as endophytes from the root of *Dendrobium nobile* [87]. The type specimens of *Xylaria venosula* is absent. Our asexual morph coincides with the description for *Xylaria* [45]. In the present study, it was isolated from roots and stems of *Dendrobium fimbriatum*, *D. primulinum*, and *Dendrobium* sp. *Xylaria venosula*, *X. arbuscula* and *X. apiculate* formed a monophyletic group with robust support (100% ML/1.0BPP). *Xylaria arbuscula* (CBS 126415 and CBS 126416) were isolated from the branch of Quercus and Robinia in a fern greenhouse (Brandenburg, Germany), respectively [81]. *Xylaria apiculata* EF6 as an endolichenic strain was isolated and identified with the ITS sequence from the lichen *Pyxine petricola* in India [94]. All nine strains in this group have very similar sequences of each gene with only 0.35–2.5% base pair differences. Few morphological information is available for comparison. Therefore, we regard all of them as *Xylaria venosula*.

## 4. Discussion

### 4.1. Xylariaceous Endophyte Associated with Dendrobium

From this study, each *Dendrobium* species is associated with two different Xylarialean species on average (including the unidentified taxa). The actual numbers would be higher considering the unidentified hosts. *Xylaria* is the most frequently isolated Xylariaceous genera associated with *Dendrobium*, followed by *Hypoxylon* and *Nemania*. *Hypoxylon* and *Xylaria* species account for the majority (29% and 51%) of all xylariaceous species. *Hypoxylon* and *Xylaria* endophytes mostly occurred on angiosperms, and a few gymnosperms were mainly distributed in pantropical areas [23,101,102,103,104]. *Hypoxylon investiens*, *Nemania diffusa*, *Nigrospora species*, *Xylaria cubensis*, *Xylaria curta*, *X. feejeensis*, *X. grammica*, and *X. venosula* are common fungal endophytes in orchids, and most of them were identified by molecular analysis [35,36,37,105,106,107,108]. *Hypoxylon officinalis* and *Xylaria venosula* were isolated from both China and Thailand. Three new hosts were recorded for *Xylaria venosula* in this study. It is worth noting that five novel taxa with one cryptic species (*Hypoxylaceae* sp.) were introduced from forty-seven strains, which implied that fungal endophyte perhaps contains many unrecognized or cryptic species. Many xylarialean endophytes from various hosts in the USA have been inferred as novel species with multi-locus [109]. Due to all novel taxa being far away from the species of *Xylaria* sensu stricto (*X. hypoxylon* aggregate, Figure 2), we defined them as *Xylaria* sensu lato.

Despite the limited sample size and scope, we speculate that *Dendrobium* species are associated with highly diverse Xylariales, especially the *Hypoxylon* and *Xylaria* species. The result coincides with that of Chen et al. (2013) who investigated 217 xylariaceous strains from seven medical *Dendrobium* species with multi-gene phylogenetic analysis in China. *Xylaria grammica* is the most isolated taxon from the roots of *Dendrobium* in southwestern China [37]. Our results showed that the penzigioid species *Xylaria berteri* was repeatedly isolated from various *Dendrobium* organs in northern Thailand, while most Hypoxylaceae strains were from Chinese *Dendrobium*. As an endophyte, *Xylaria berteri* has often been recorded in tropical areas such as Brazil, Hawaiian Islands, Mexico, and Panama [110,111,112]. *Hypoxylon officinalis* was introduced based on the traits of three strains from roots of different *Dendrobium*, so we speculated *Hypoxylon officinalis* could be an organ specificity species. However, the organ, climatic or geographic preference, and host specificity in Xylariales endophyte needs further investigation in a wide range [48].

### 4.2. The Dilemma for Xylaria Endophyte Verification

Xylariaceae has been revised several times [19,23]. The best investigated *Xylaria* combines the traditional morphological concepts based on the type of material with multi-gene sequence analysis [16,23]. However, many old but well-defined *Xylaria* species lack type material and sequence data [23,48]. Few reports relate to the *Xylaria* asexual morph. In addition, a considerable number of *Xylaria* species produce sterile mycelia only, and reproduction rarely occurs on artificial media [37,78]. Altogether these make the *Xylaria* endophyte species resolution very difficult [23,45,109,113]. Therefore, a multi-locus phylogenetic analysis is the most reliable approach to know the taxonomic placement of the *Xylaria* endophyte [37]. Some cryptic *Xylaria* species have been found [37,102,109]. In this study, most *Xylaria* species identification mainly depends on molecular phylogenetic analysis. We could only present limited morphological characterization with asexual morph sporulated on 4% OA and little available sexual morph record for comparison. However, the discrimination based on base pairs could result from pseudogene or over-evaluation of genes [114]. With the type specimen re-inspection, assignment, and phylogenomic deciphering, we may approach a well-supported taxonomy for Xylariales [114,115,116]. Therefore, all novelties in this study could be regarded as tentative new taxa. Besides type (and ex-type) assignment, sporulation for endophytes such as *Xylaria* is always encouraged [42]. More attention should be paid to *Xylaria* asexual morph, which links endophytes with the sexual morph of known species and contributes to deciphering its genealogy with more novel, even cryptic species [54]. A breakthrough is needed for *Xylaria* asexual morph study, and any pertaining characteristics should be recorded for recognition, comparison, and discussion even though they would be criticized and potentially wrong. We recommend using OA with low concentrations under alternative fluorescent light to induce sporulation of *Xylaria* endophytes as more and more studies successfully sporulate *Xylaria* species with this medium [45,111]. The three new species, *Xylaria insolita*, *X. necrophora*, and *X. subescharoidea*, have been introduced recently with asexual morph sporulated on OA [117,118]. Due to a lack of full information for type species, we did not designate any novel *Xylaria* species in this study.

### 4.3. HTS Application for Fungal Endophyte

Compared with our results, the high-throughput amplicon sequencing (HTS) implemented by ITS-rRNA metagenomics analysis for the fungal endophyte composition of *Dendrobium officinale* showed a very low *Xylaria* frequency, which perhaps is due to the designed primers not being enough for all endophytes [119]. Although multi-gene markers have been employed for further resolving many taxonomic groups, including fungal endophytes, the traditional DNA-sequencing method only allows individual specimen sequencing [120]. The next generation sequencing (NGS) makes it possible to sequence mixed environmental bulk samples [121]. The collection of the resulting set of tens of thousands of genetic reads in parallel contributes to understanding historical, functional, and ecological biodiversity [2,122]. HTS application to fungal endophytes reveals some sympatric cryptic species, which discovered taxa that could not grow on artificial media [118,123,124]. Meanwhile, the HTS often produces false negative or erroneous results, and its application normally requires multiple barcoding reads into operational taxonomic units (OTUs), which is feasible to discern environmental DNA to generic levels but loses information on intraspecific diversity [122]. Therefore, applying the culture-dependent method or HTS only maybe not reveal the real composition of fungal endophytes. It is likely that combining both methods or improving the HTS primers can contribute to discovering the real composition, taxonomic placements, and ecological strategies [125,126,127]. Several advanced metabarcoding strategies and platforms have been introduced to achieve species identification in dietary, gut microbiome, and wildlife forensic species detection, which is likely to be used for more effective and exact fungal endophytes identification [128].

### 4.4. Potential Roles of Xylariaceous Endophyte

In this study, several species were initially identified as pathogens reported on other plants. Most grasses and reeds perhaps harbor species of *Apiospora* endophyte [65]. *Apiospora* (=*Arthrinium*) sp. was also associated with *Dendrobium candidum* and *D. nobile* [36,90]. *Nigrospora sphaerica* is an opportunistic pathogen causing onychomycosis in humans and can cause leaf blight on *Camellia sinensis* [55,129,130]. *Nigrospora chinensis* is a common pathogen that has been reported from *Ginkgo biloba* [90]. *Nigrospora* as ubiquitous endophyte, plant, and human pathogens, have been found in several orchids including *Bulbophyllum neilgherense*, *Dendrobium candidum*, and *Vanda testacea* [90,131]. The occurrence of the appressoria-like structure found in the 4% of OA culture of *Xylaria venosula* suggests this species could be invasive in the germination period. These fungal endophytes probably become potential pathogens or saprobes involving senescence during later life [25,132,133,134]. It is reported that some volatile organic compounds isolated from Xylariales species can cause pest larval mortality [88,130,135]. For further understanding of xylarialean endophyte, an optimized methodology is needed on the taxonomy, physiology, and functional roles.

## 5. Conclusions

*Dendrobium* are associated with various Xylariales species. *Hypoxylon* and *Xylaria* take the majority of culturable xylarialean fungal endophytes in *Dendrobium* species. *Hypoxylon endophyticum*, *Xylaria*
*berteri* and *X. grammica* are the most frequently culture-dependent isolates. Novel endophytic species need robust backbone phylogenetic analysis with complete type material information. The low concentration of OA is recommended for inducing sporulation of *Xylaria* endophytes.

## Figures and Tables

**Figure 1 jof-08-00248-f001:**
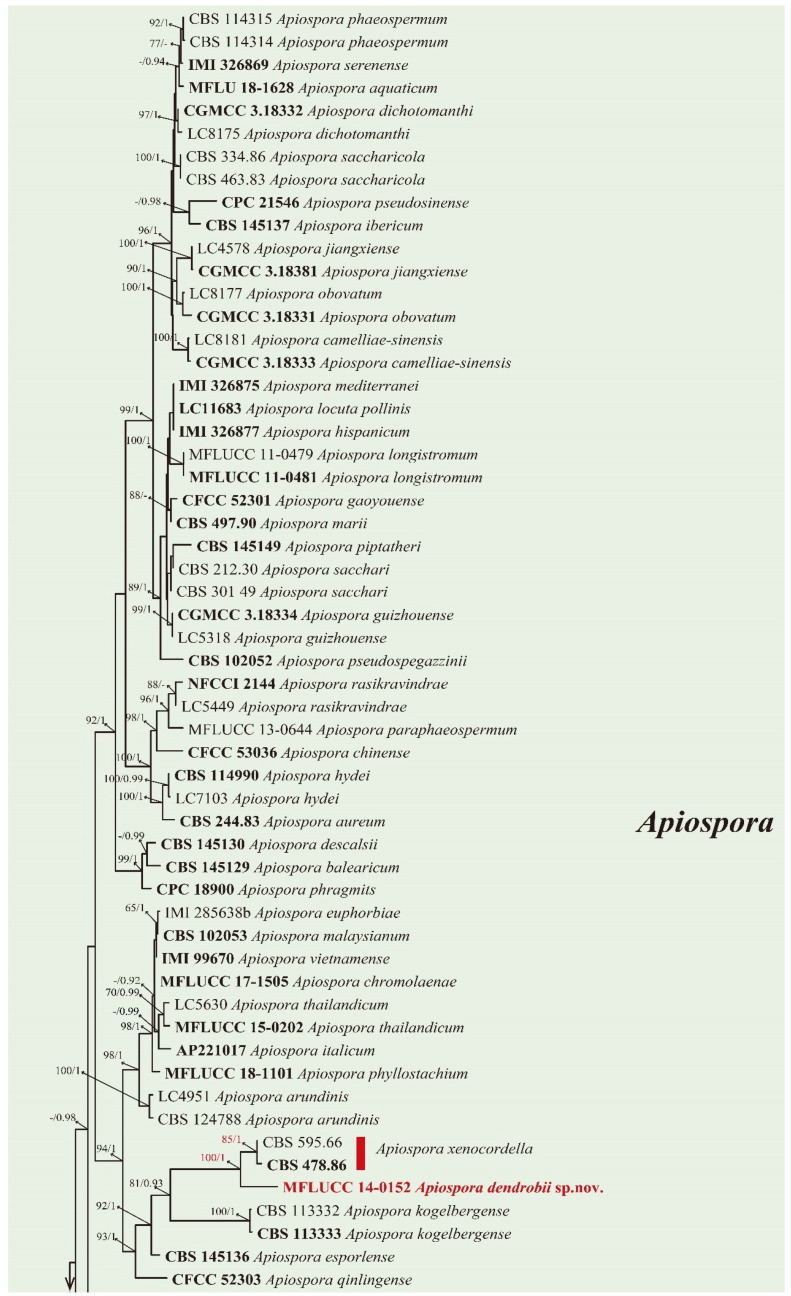
A multi-locus phylogenetic tree based on the combined ITS-LSU-TUB2-TEF-1α sequences of *Apiosporaceae* species resulting from a maximum likelihood analysis. *Pseudomassaria chondrospora* (MFLUCC 14-0545), *Pseudomassaria sepincoliformis* (CBS 129022), and *Pseudopiospora corni* (MFLUCC 14-0544) are selected as outgroup. The isolates from this study are in red. The ex-type isolates are in bold. Maximum likelihood values (ML) equal or greater than 70 and/or 0.9 Bayesian posterior probabilities (BPP) are labelled at the end of nodes. Dashes are indicated values lower than 70% ML and/or 0.9BPP. Scale bar corresponds to 0.09 substitutions per site.

**Figure 2 jof-08-00248-f002:**
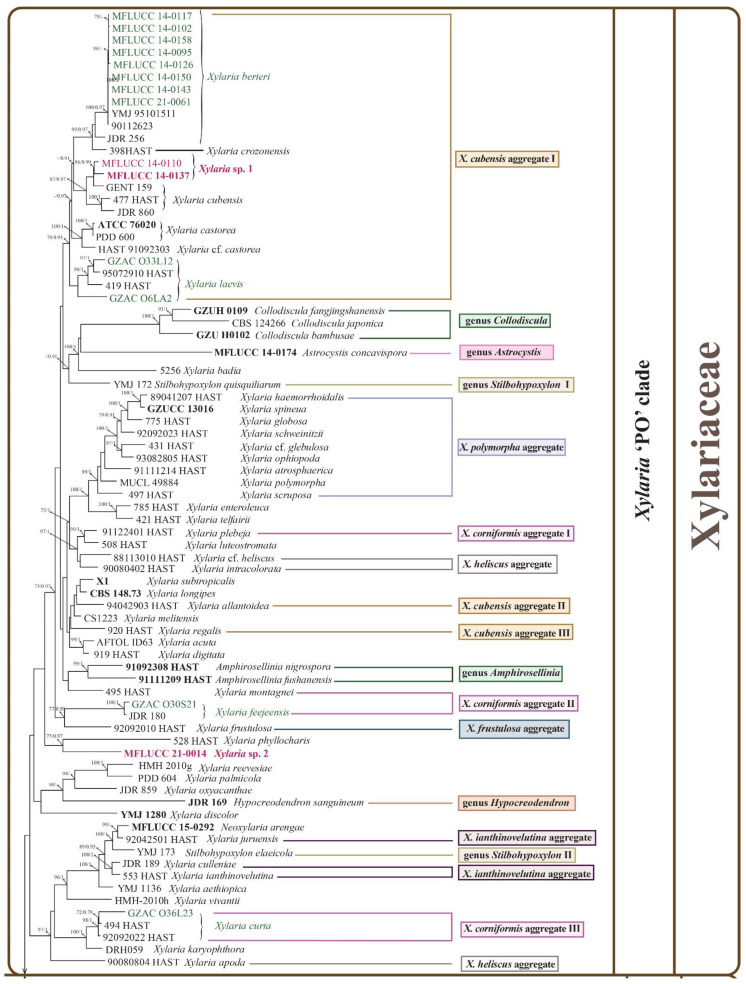
A multi-locus phylogenetic tree based on the combined ITS-LSU-TUB2-RPB2 sequences of related Xylariales species resulting from a maximum likelihood phylogenetic analysis. The tree is rooted at the outgroup *Lopadostoma gastrinum* (CBS 134632) and *Lopadostoma insulare* (CBS 133214). The isolates from this study are in red (tentative new and new species) and green (known species). Strains needing reevaluation are in purple. Different aggregate partition is based on their various stromata, mainly represented by several *Xylaria* species following Hsieh et al. 2010. The ex-type isolates are in bold. Equal or greater than 70% maximum likelihood values (ML) and/or 0.9 Bayesian posterior probabilities (BPP) are labelled at the end of nodes. Dashes are indicated values lower than 70% ML and/or 0.9BPP. Scale bar corresponds to 0.09 substitutions per site.

**Figure 3 jof-08-00248-f003:**
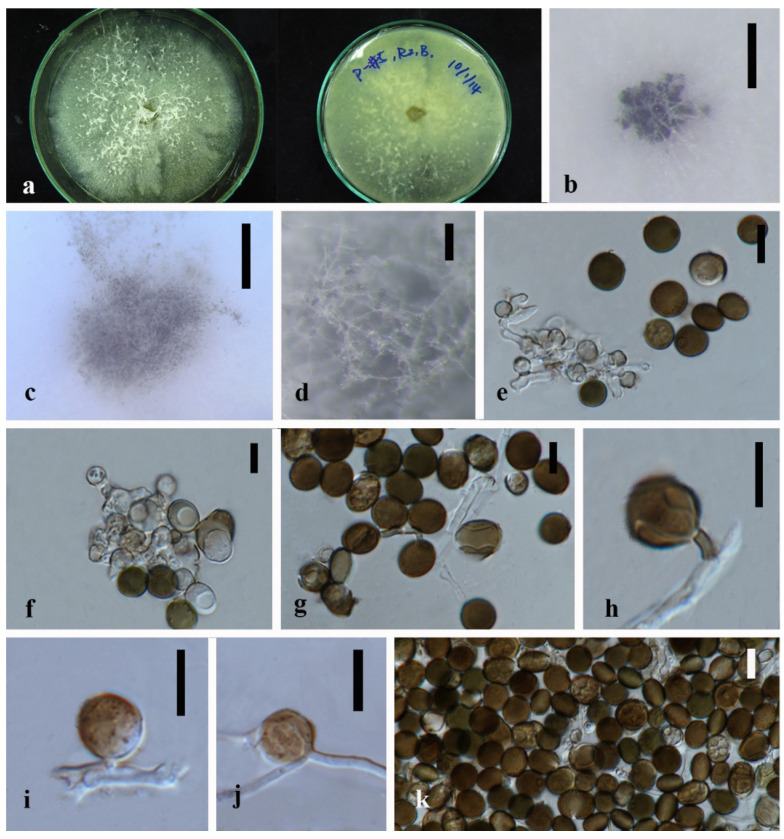
*Apiospora dendrobii* (MFLU 21-0153). (**a**) Colony on PDA (left-front view, right-reverse view). (**b**–**d**) Conidiomatal mass. (**e**–**g**) Conidiophores with conidia. (**h**–**j**) Chlamydospores. (**k**) Conidia. Notes: (**b**–**k**) on 4% OA. Scale bars: (**b**) = 100 μm, (**e**–**k**) = 10 μm.

**Figure 4 jof-08-00248-f004:**
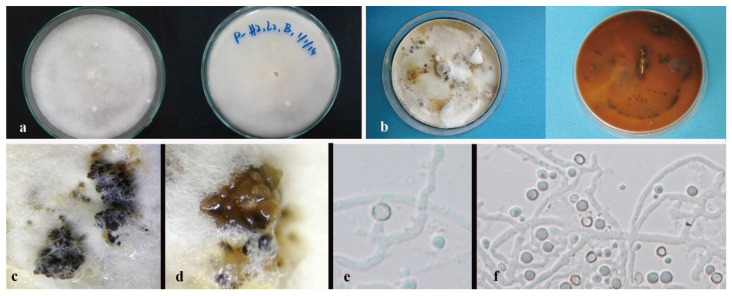
*Nigrospora chinensis* (MFLUCC 14–0109). (**a**) Colony on PDA (left-front view, right-reverse view). (**b**) Colony on MEA. (**c**,**d**) Conidiomatal masses. (**e**) Mycelia. (**f**) Conidia. Notes: (**c**–**f**) on 2% MEA.

**Figure 5 jof-08-00248-f005:**
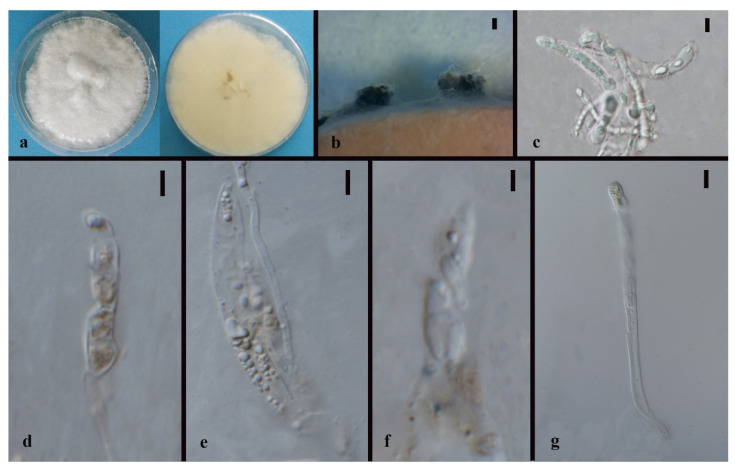
*Annulohypoxylon moniliformis* (MFLU-O35L22) (**a**) Colony on PDA (left-front view, right-reverse view). (**b**) Ascomata on WA (slide cultre). (**c**–**g**) Asci with ascospores on the robber band (for supporting slide) in slide culture. Scale bars: (**b**) = 200 μm, (**c**–**g**) = 10 μm.

**Figure 6 jof-08-00248-f006:**
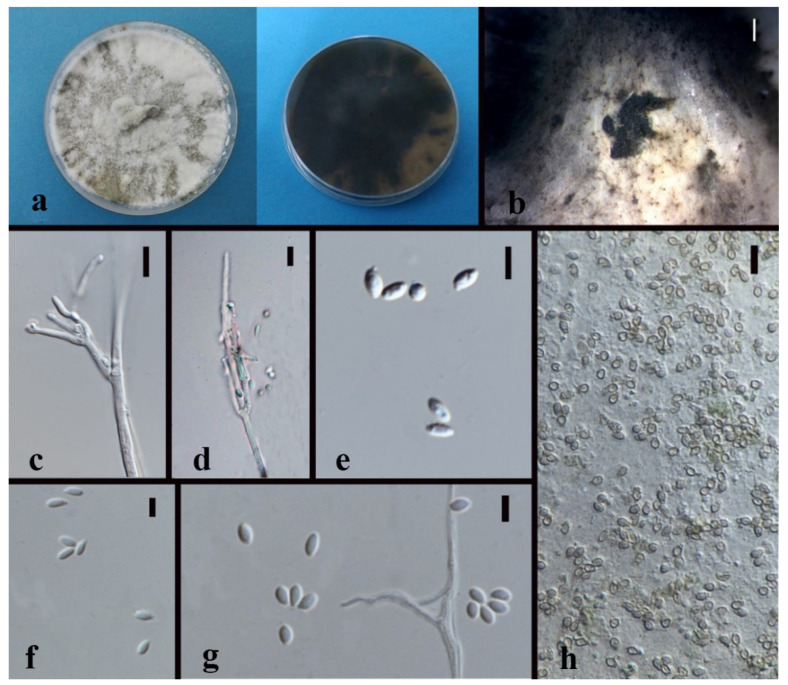
*Hypoxylon endophyticum* (MFLU 21-0154) (**a**) Colony on PDA (left-front view, right-reverse view). (**b**) Conidiomata. (**c**,**d**) Conidiophores with conidia. (**e**–**h**) Conidia. Notes: (**b**–**g**) on WA, slide culture, (**h**) on PDA. Scale bars: (**b**) = 200 μm, (**c**,**d**) = 10 μm, (**e**–**g**) = 5 μm, (**h**) = 10 μm.

**Figure 7 jof-08-00248-f007:**
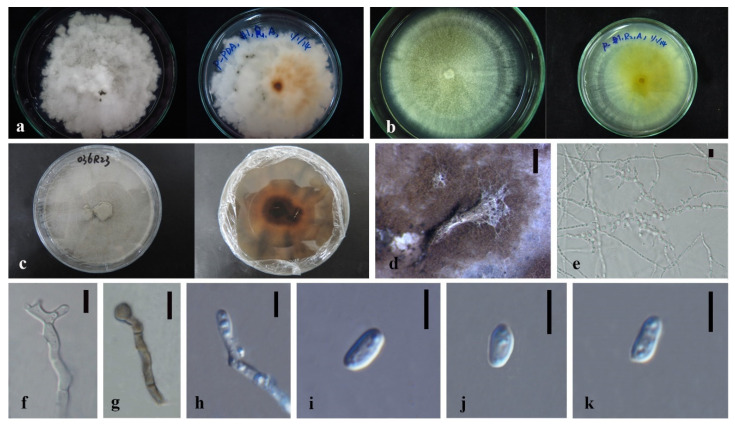
*Hypoxylon officinalis* (MFLU 21-0152) (**a**) Colony on PDA (MFLUCC 14-0075). (**b**) Colony on PDA (MFLUCC 14-0078). (**c**) Colony on PDA (MFLUCC 21-0060). (**d**) Conidiomata. (**e**) Mycelia. (**f**–**h**) Conidiophores with conidia. (**i**–**k**) Conidia. Notes: (**d**,**f**–**g**) on MEA, (**e**) on WA, (**h**–**k**) on 4% OA. Scale bars: (**d**) = 500 μm, (**e**–**k**) = 5 μm.

**Figure 8 jof-08-00248-f008:**
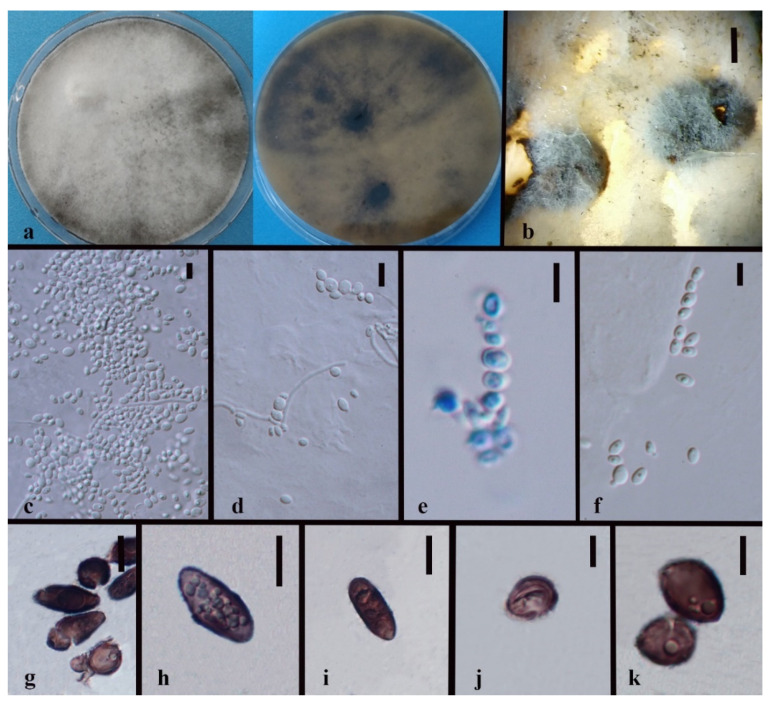
*Hypoxylon investiens* (MFLUCC 15-1155) (**a**) Colony on PDA (left-front view, right-reverse view). (**b**) Aggregated conidiomata. (**c**–**f**) Conidia. (**g**–**k**) Ascospores. Notes: (**b**) on PDA, (**c**–**f**) on WA (slide culture), (**g**–**k**) on MEA. Scale bars: (**b**) = 1 cm, (**c**–**f**) = 5 μm, (**g**–**k**) = 10 μm.

**Figure 9 jof-08-00248-f009:**
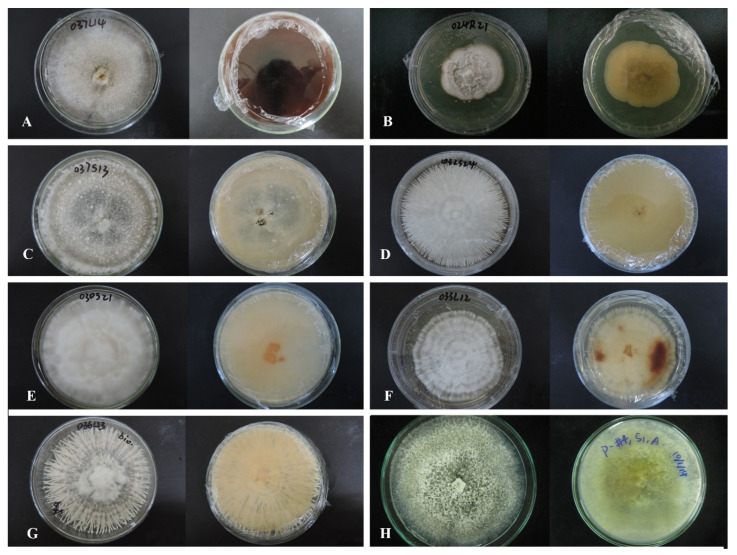
Front and reverse views of the isolates on PDA, isolated *Dendrobium* species; (**A**) *Hypoxylon pulicicidum* (GZAC O37L14). (**B**) *Induratia* sp. (MFLUCC 15-1218). (**C**) *Nigrospora sphaerica* (GZAC O37S13). (**D**) *Xylaria papulis* (GZAC O32S24). (**E**) *Xylaria feejeensis* (GZACA O30S21). (**F**) *Xylaria laevis* (GZACA O33L12). (**G**) *Xylaria curta* (GZAC O36L23). (**H**) *Xylaria berteri* (MFLUCC 14-0142).

**Figure 10 jof-08-00248-f010:**
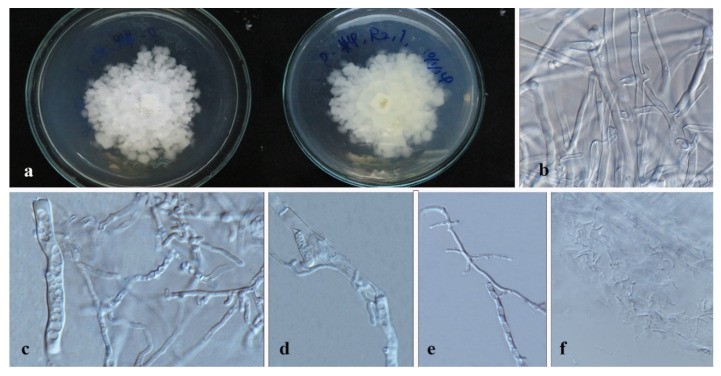
*Hypoxylaceae* sp. (**a**) Colony on PDA (MFLUCC 14-0141) (left-front view, right-reverse view). (**b**,**c**) Mycelia. (**d**,**e**) Conidiophore-like structures. (**f**) β conidia-like structures Notes: (**b**–**f**) on 4% OA.

**Figure 11 jof-08-00248-f011:**
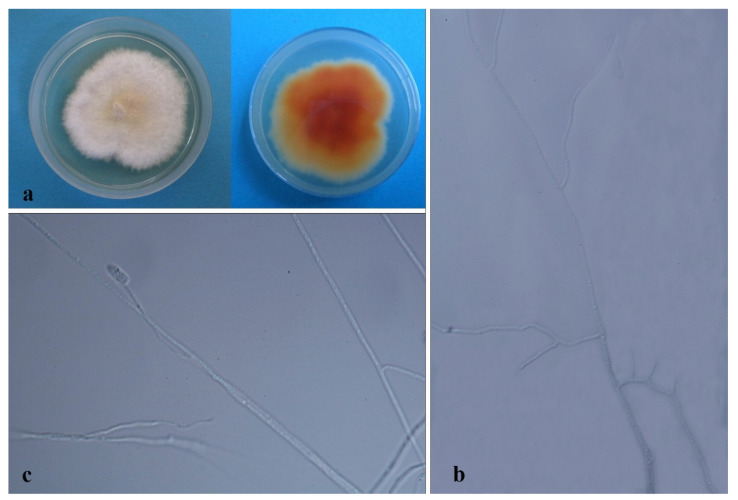
*Nemania dendrobii* (GZAC O49S1A) (**a**) Colony on PDA (MFLUCC 18-1213) (left-front view, right-reverse view). (**b**) Mycelia. (**c**) Conidiophore with conidia. Notes: (**b**,**c**) on WA.

**Figure 12 jof-08-00248-f012:**
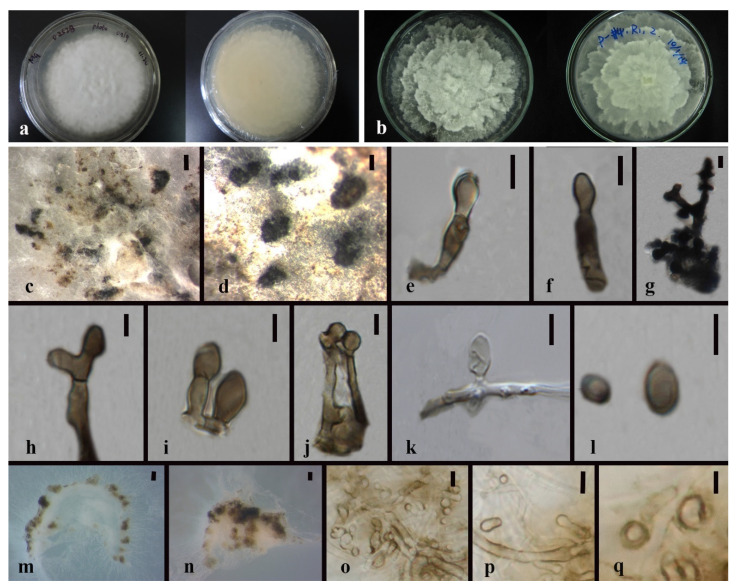
*Nemania bipapillata* (MFLUCC 14-0105). (**a**) Colony on PDA (MFLUCC 14-0105) (left-front view, right-reverse view). (**b**) Colony on PDA (MFLUCC 14-0138) (left-front view, right-reverse view). (**c**,**d**) Conidiomata. (**e**,**f**), (**h**–**j**) Conidiophores with conidia. (**g**) Stromatic hypha with protuberances. (**k**) Chlamydospores. (**l**) Conidia. (**m**,**n**) Conidiomatal masses. (**o**–**q**) Conidia. Notes: (**c**–**l**) on PDA, (**m**–**q**) on WA. Scale bars: (**c**,**d**) = 200 μm, (**e**–**l**) = 5 μm, (**m**,**n**) = 500 μm, (**o**) = 10 μm, (**p**,**q**) = 5 μm.

**Figure 13 jof-08-00248-f013:**
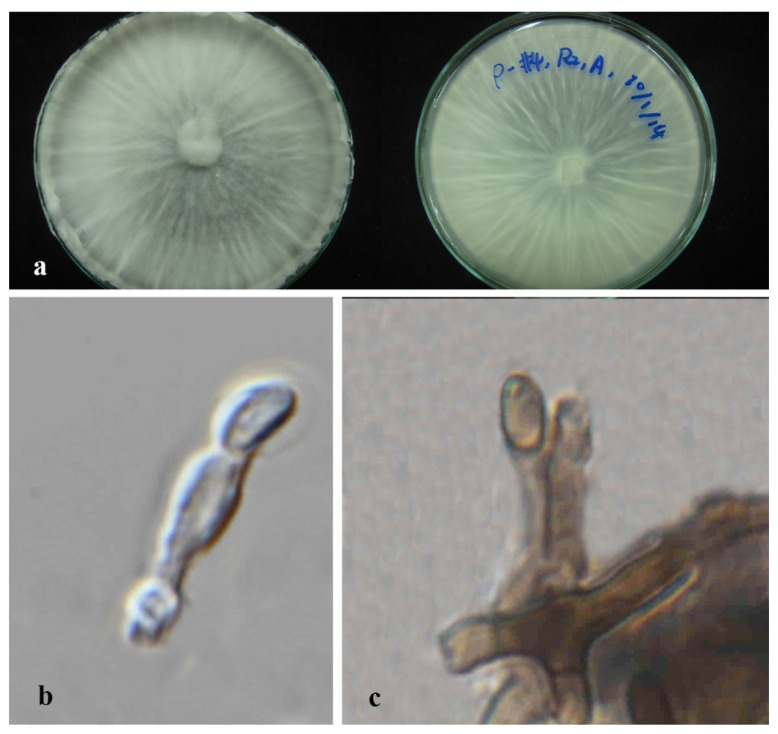
*Nemania diffusa* (MFLUCC 14-0139) (**a**) Colony on PDA (left-front view, right-reverse view). (**b**,**c**) Conidiophores with conidia-like structures.

**Figure 14 jof-08-00248-f014:**
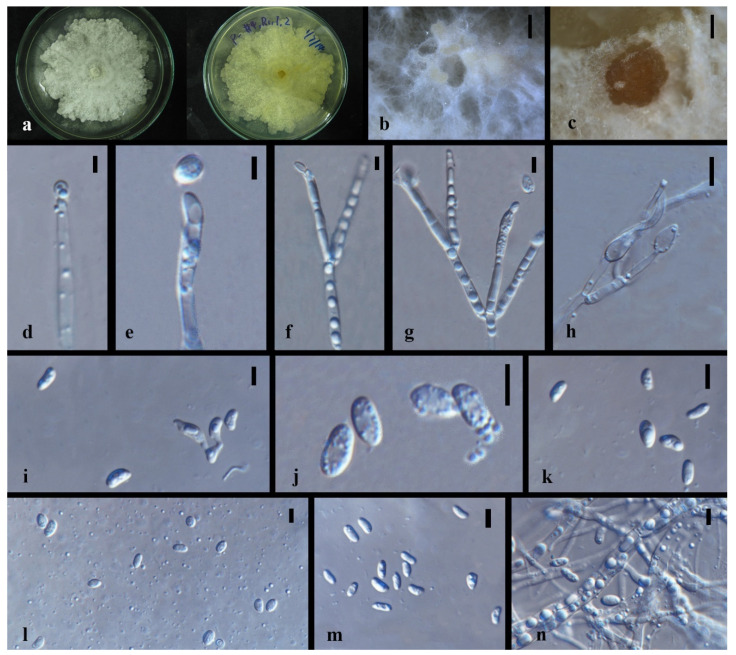
*Xylaria* sp. 1 (**a**) Colony on PDA (MFLUCC 14–0137) (left-front view, right-reverse view). (**b**,**c**) Conidiomata. (**d**–**h**) Conidiophores with conidia. (**i**–**m**) Conidia. n Mature mycelia. Notes: (**b**–**n**) on 4% OA. Scale bars: (**b**,**c**) = 500 μm, (**d**–**n**) = 5 μm.

**Figure 15 jof-08-00248-f015:**
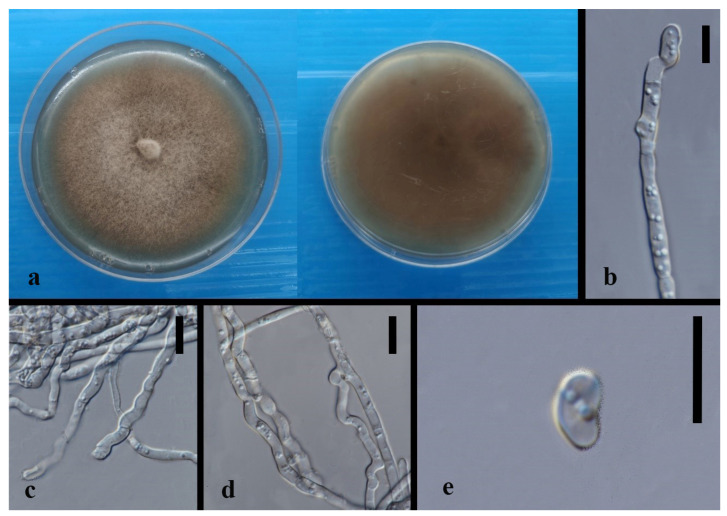
Xylaria sp.2. (**a**) Colony on 2% MEA (MFLUCC 21-0014) (left-front view, right-reverse view). (**b**) Conidiophore with conidium. (**c**,**d**) Mycelia. (**e**) Conidium. Scale bars: (**b**–**e**) = 5 μm. Notes: (**b**–**e**) on 4% OA.

**Figure 16 jof-08-00248-f016:**
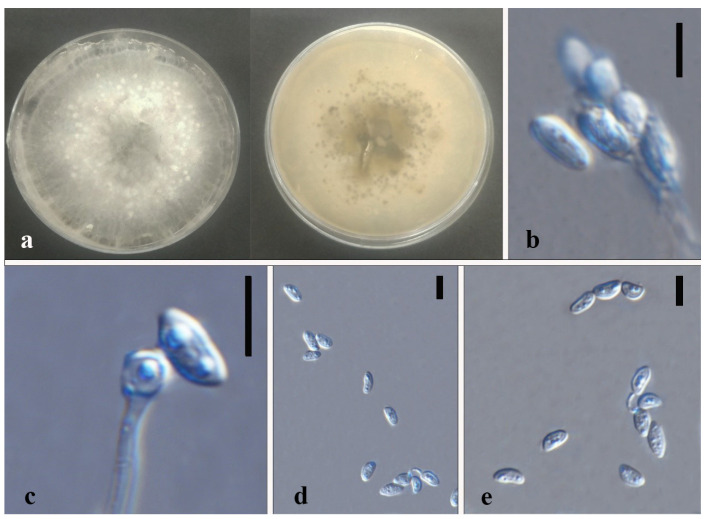
*Xylaria* sp. 3 (MFLUCC 21-0059). (**a**) Colony on 4% OA (MFLUCC 21-0059) (left-front view, right-reverse view). (**b**,**c**) Conidia with conidiophores. (**d**,**e**) Conidia. Notes: (**b**–**e**) on 4% OA. Scale bars: (**b**–**e**) = 5 μm.

**Figure 17 jof-08-00248-f017:**
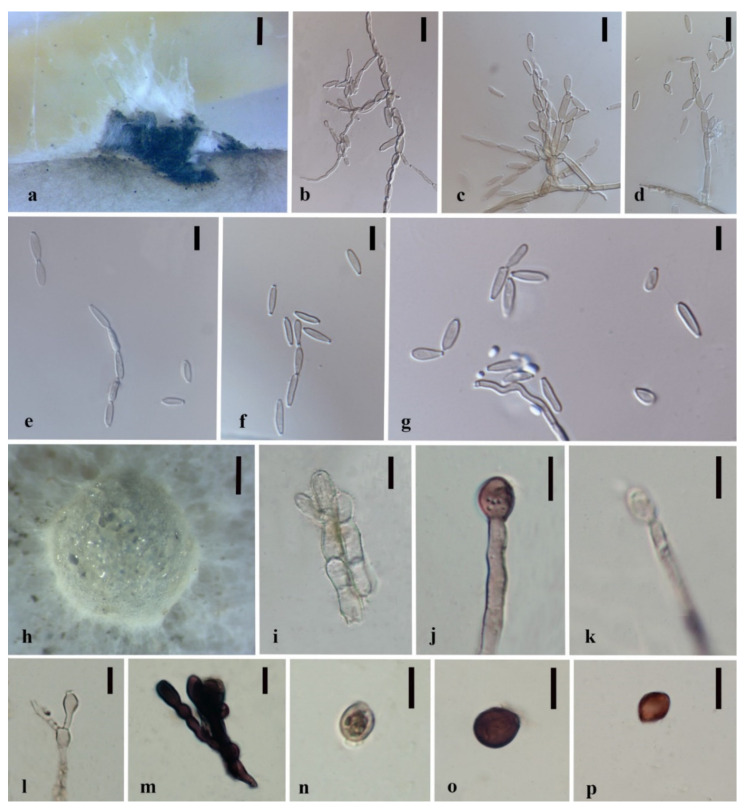
*Xylaria berteri*. (**a**) Conidiomata. (**b**–**d**) Conidiophores with conidia. (**e**–**g**) Conidia. (**h**) Conidiomata. (**i**–**m**) Conidiophore-like structures. (**n**–**p**) Conidia-like structures. Notes: (**a**–**g**) on WA, slide culture, (**h**–**p**) on PDA. Scale bars: (**a**) = 200 μm, (**b**–**g**) = 10 μm, (**h**) = 500 μm, (**i**–**p**) = 10 μm.

**Figure 18 jof-08-00248-f018:**
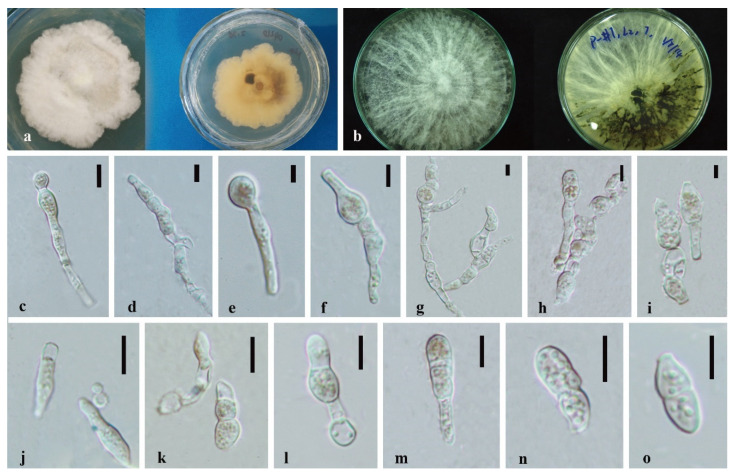
*Xylaria grammica* on PDA and WA (slide culture). (**a**) MFLUCC 14-0093 colony on PDA (left-front view, right-reverse view). (**b**) MFLUCC 14-0146 colony on PDA (left-front view, right-reverse view). (**c**–**i**) Conidiophores with conidia-like structures. (**j**–**o**) Conidia-like structures. Scale bars: (**c**–**i**) = 5 μm, (**j**–**o**) = 10 μm. Note: (**c**–**o**) on WA of slide culture.

**Figure 19 jof-08-00248-f019:**
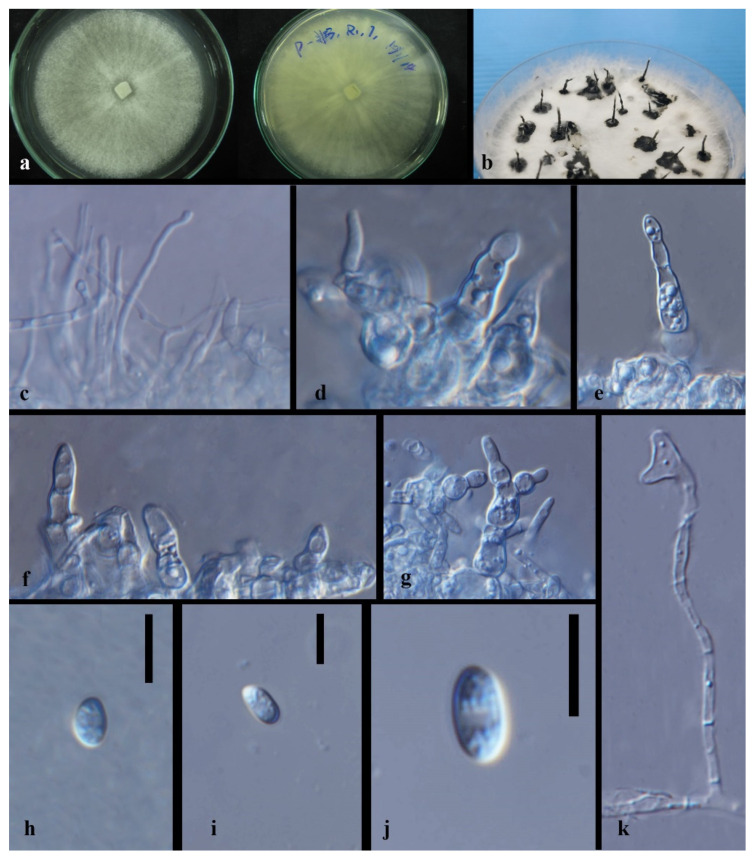
*Xylaria venosula*. (**a**) Coloy on PDA (MFLUCC 14-0114) (left-front view, right-reverse view). (**b**) Stroma on 2% MEA. (**c**–**g**) Conidiophores with conidia. (**h**–**j**) Conidia. (**k**) Appressoria-like structure. Notes: (**c**–**k**) on 4% OA. Scale bars: (**h**–**j**) = 5 μm.

**Table 1 jof-08-00248-t001:** Information on *Dendrobium* orchids from this study.

Code	Sampling Site	*Dendrobium* Species
1	Animal Husbandry and Veterinary Institute, Guiyang, Guizhou, China	*Dendrobium chrysanthum* Lindl.
2	Orchid nursery 1, Luodian, Guizhou, China	*Dendrobium officinale* Kimura & Migo
3	Orchid nursery 2, Luodian, Guizhou, China	*Dendrobium aphyllum* (Roxb.) C. E.
		*Dendrobium aurantiacum* Rchb. f. var. *denneanum*
		*Dendrobium chrysotoxum* Lindl.
		*Dendrobium fimbriatum* Hook.
		*Dendrobium hercoglossum* Rchb. f.
		*Dendrobium moniliforme* (L.) Sw.
		*Dendrobium officinale* Kimura & Migo
		*Dendrobium primulinum* Lindl.
4	Orchid nursery, Xingyi, Guizhou, China	*Dendrobium loddigesii* Rolfe
		*Dendrobium officinale* Kimura & Migo
		*Dendrobium* sp. 5
		*Dendrobium* sp. 6
		*Dendrobium* sp. 7
		*Dendrobium* sp. 8
5	Steep forest, Doitung, Mae Fah Luang District, Chiang Rai, Thailand	*Dendrobium* sp. 4
6	Wat Phra That Doi Tung (Temple of Doi Tung Pagodas), Mae Fah Luang District, Chiang Rai, Thailand	*Dendrobium cariniferum* Rchb. f.
		*Dendrobium harveyanum* Rchb.f.
		*Dendrobium moschatum* (Buch.Ham.) Sw
		*Dendrobium* sp. 1
		*Dendrobium* sp. 2
		*Dendrobium* sp. 3

**Table 2 jof-08-00248-t002:** PCR thermal cycling protocols.

Region/Gene	Primers	Cycle Number	Denaturation	Annealing	Elongation	Reference
ITS	ITS1/ITS4	30	95 °C 1 min	53 °C 1 min	72 °C 1 min	White et al., 1990
LSU	LR0R/LR5	35	94 °C 1 min	53 °C 50 s	72 °C 1 min 30 s	Vilgalys and Hester 1990
RPB2	Frpb-5f/frpb-7cr	35	95 °C 45 s	57 °C 50 s	72 °C 1 min 30 s	Liu et al., 1999
TEF-1α	728F/986R	36	95 °C 45 s	56 °C 1 min	72 °C 1 min	Carbone and Kohn 1999
TUB2	Bt2A/Bt2B	35	94 °C 1 min	55 °C 55 s	72 °C 1 min	Glass and Donaldson 1995

**Table 3 jof-08-00248-t003:** Endophytic Xylariaceous isolates from *Dendrobium* orchids in this study.

Xylariales Species	Strain Code	Host *Dendrobium* Orchids	Tissues
*Annulohypoxylon moniliformis* sp. nov.	MFLUCC 18-1214 ^HT^	*Dendrobium moniliforme*	Leaf
*Apiospora dendrobii* sp. nov.	MFLUCC 14-0152 ^HT^	*Dendrobium harveyanum*	Root
*Hypoxylon endophyticum* sp. nov.	MFLUCC 18-1206 ^HT^	*Dendrobium aphyllum*	Root
*Hypoxylon endophyticum* sp. nov.	MFLUCC 18-1209	*Dendrobium huoshanense*	Stem
*Hypoxylon endophyticum* sp. nov.	MFLUCC 18-1211	*Dendrobium chrysotoxum*	Leaf
*Hypoxylon endophyticum* sp. nov.	MFLUCC 18-1208	*Dendrobium* sp. 5	Root
*Hypoxylon endophyticum* sp. nov.	MFLUCC 18-1210	*Dendrobium loddigesii*	Root
*Hypoxylon endophyticum* sp. nov.	MFLUCC 18-1207	*Dendrobium hercoglossum*	Leaf
*Hypoxylon officinalis* sp. nov.	MFLUCC 14-0075 ^HT^	*Dendrobium* sp. 1	Root
*Hypoxylon officinalis* sp. nov.	MFLUCC 14-0078	*Dendrobium* sp. 1	Root
*Hypoxylon officinalis* sp. nov.	MFLUCC 21-0060	*Dendrobium officinale*	Root
*Hypoxylon investiens* (Schwein.) M.A. Curtis	MFLUCC 15-1155	*Dendrobium moschatum **	Stem
*Hypoxylon pulicicidum* J. Fournier, Polishook & Bills	GZAC O37S13	*Dendrobium hercoglossum **	Root
*Hypoxylaceae* sp.	MFLUCC 14-0141	*Dendrobium* sp. 3	Root
*Induratia* sp.	MFLUC C 15-1218	*Dendrobium* sp. 4	Stem
*Nemania dendrobii* sp. nov.	MFLUCC 18-1213 ^HT^	*Dendrobium* sp. 7	Stem
*Nemania dendrobii* sp. nov.	MFLUCC 18-1212	*Dendrobium* sp. 6	Root
*Nemania diffusa* (Sowerby) Gray	MFLUCC 14-0139	*Dendrobium* sp. 2	Root
*Nemania bipapillata* (Berk. & M.A. Curtis) Pouzar	MFLUCC 14-0138	*Dendrobium* sp. 3	Root
*Nemania bipapillata* (Berk. & M.A. Curtis) Pouzar	MFLUC C 14-0105	*Dendrobium cariniferum **	Stem
*Nigrospora chinensis* Mei Wang & L. Cai	MFLUCC 14-0109	*Dendrobium cariniferum **	Stem
*Nigrospora chinensis* Mei Wang & L. Cai	MFLUCC 18-1215	*Dendrobium officinale*	Root
*Nigrospora sphaerica* (Sacc.) E.W. Mason	GZAC O37S13	*Dendrobium hercoglossum **	Leaf
*Xylaria berteri* (Mont.) Cooke ex J.D. Rogers & Y.M. Ju	MFLUCC 14-0095	*Dendrobium cariniferum **	Root
*Xylaria berteri* (Mont.) Cooke ex J.D. Rogers & Y.M. Ju	MFLUCC 14-0102	*Dendrobium cariniferum*	Stem
*Xylaria berteri* (Mont.) Cooke ex J.D. Rogers & Y.M. Ju	MFLUCC 14-0143	*Dendrobium* sp.	Stem
*Xylaria berteri* (Mont.) Cooke ex J.D. Rogers & Y.M. Ju	MFLUCC 14-0117	*Dendrobium* sp. 2	Root
*Xylaria berteri* (Mont.) Cooke ex J.D. Rogers & Y.M. Ju	MFLUCC 14-0126	*Dendrobium* sp. 2	Leaf
*Xylaria berteri* (Mont.) Cooke ex J.D. Rogers & Y.M. Ju	MFLU CC 14-0143	*Dendrobium* sp. 3	Stem
*Xylaria berteri* (Mont.) Cooke ex J.D. Rogers & Y.M. Ju	MFLUCC 14-0150	*Dendrobium harveyanum **	Root
*Xylaria berteri* (Mont.) Cooke ex J.D. Rogers & Y.M. Ju	MFLUCC 14-0158	*Dendrobium harveyanum*	Leaf
*Xylaria berteri* (Mont.) Cooke ex J.D. Rogers & Y.M. Ju	MFLUCC 21-0061	*Dendrobium* sp. 2	Leaf
*Xylaria curta* Fr.	GZAC O36L23	*Dendrobium officinale*	Leaf
*Xylaria feejeensis* (Berk.) Fr.	GZAC O30S21	*Dendrobium aphyllum **	Stem
*Xylaria grammica* (Mont.) Mont.	MFLUCC 14-0093	*Dendrobium* sp. 1	Leaf
*Xylaria grammica* (Mont.) Mont.	MFLUCC 14-0146	*Dendrobium* sp. 3	Leaf
*Xylaria laevis* Lloyd	GZAC O33L12	*Dendrobium aurantiacum* var. *denneanum **	Leaf
*Xylaria laevis* Lloyd	GZAC O6LA2	*Dendrobium officinale*	Leaf
*Xylaria papulis* Lloyd	GZAC O32S24	*Dendrobium chrysotoxum*	Stem
*Xylaria* sp.1	MFLUCC 14-0137	*Dendrobium* sp. 3	Root
*Xylaria* sp.1	MFLUCC 14-0110	*Dendrobium cariniferum*	Leaf
*Xylaria* sp.2	MFLUCC 21-0014	*Dendrobium chrysanthum*	Stem
*Xylaria* sp.3	MFLUCC 21-0059	*Dendrobium aphyllum*	Root
*Xylaria venosula* Speg.	MFLUCC 14-0114	*Dendrobium* sp. 2	Root
*Xylaria venosula* Speg.	MFLUCC 21-0013	*Dendrobium fimbriatum **	Root
*Xylaria venosula* Speg.	MFLUCC 21-0015	*Dendrobium aurantiacum* var. *denneanum **	Root
*Xylaria venosula* Speg.	MFLUCC 21-0016	*Dendrobium primulinum **	Stem
*Xylaria venosula* Speg.	MFLUCC 21-0017	*Dendrobium* sp. 8	Stem

Note: Ex-type strains are in bold. HT are abbreviations of holotype. The label ‘HT’ means that their dry cultures are designed as holotype herbarium. * Is labeled at the right upper conner of host species indicates new host record.

**Table 4 jof-08-00248-t004:** Comparison of the (tentative) novel species with the closest related taxa.

New Species Names	Type Species	Morphological Differences	Phylogenetic Differences (Exclude Gaps)	References
*Annulohypoxylon moniliformis* MFLUCC 18-1214	*Annulohypoxylon* annulatum CBS 140775	No difference observed	Close to *Annulohypoxylon annulatum* CBS 140775 (98% ML/1.0BPP). Differ by 1.53% (13/847 bp) of ITS and 7.92% (42/530 bp) of TUB2	Ju and Rogers (1996), Sir et al. (2016)
*Apiospora dendrobii* MFLUCC 14-0152	*Apiospora xenocordella* CBS 478.86	Conidiogenous cell: straight (*A. dendrobii*) vs. verruculose, globose to clavate to doliiform (*A. xenocordella*).	Close to *Apiospora xenocordella* (100% ML/1.0BPP). Differ by 1.95% (13/666 bp) of ITS, 3.24% (7/216 bp) of TUB2	Crous and Groenewald (2013)
*Hypoxylon endophyticum* MFLUCC 18-1206	No ex-type available (Compared with *Hypoxylon investiens* YMJ 89062905)	No difference observed	Close to *Hypoxylon investiens* (100%ML/1.0BPP). Differ by 3.54% (30/847 bp) of ITS and 1.89% (15/530 bp) of TUB2	Ju and Rogers (1996), Platas et al. (2009)
	*Hypoxylon investiens* CBS 118185	No difference observed	Different lineages. Differ by 2.13% (18/847 bp) of ITS, 13.8% (198/1432 bp) of LSU, 4.93% (64/1298 bp) of RPB2 and 2.83% (15/530 bp) of TUB2	Ju and Rogers (1996), Platas et al. (2009)
*Hypoxylon officinalis* MFLUCC 14-0075	*Hypoxylon lateripigmentum* MUCL 53304	No periconiella-like conidiogenous structure was observed in *H. officinalis*	Close to *H. lateripigmentum* (100%ML/1BPP). Differ by 5.08% (43/847 bp) of ITS, 2.03% (29/1432 bp) of LSU, 8.3% (44/530 bp) of TUB2	Kuhnert et al. (2014)
*Nemania dendrobii* MFLUCC 18-1213	No ex-type available (Compared with *Nemania bipapillata* 90080610 (HAST))	No difference observed	Close to *N. bipapillata* 90080610 (100% ML/1.0BPP). Differ by 2.48% (21/847 bp) of ITS; 4.15% (22/530 bp) of TUB2, 4.93% (64/1298 bp) of RPB2	Ju and Rogers (1999), Hsieh et al. (2010), Database for Rice Seed-borne *Fungi-Nemania* sp. 15008
*Xylaria* sp.1 MFLUCC 14-0137	No ex-type available (Compared with *Xylaria cubensis* JDR 860)	In *X. cubensis*, Conidiogenous cell has round to denticulate secession scars; Conidia have flat basal abscission scar	Close to *X. cubensis*. Differ by 4.37% (37/847 bp) of ITS and 5.28% (28/530 bp) of TUB2	Rogers 1984, Rodrigues et al. (1993), Hsieh et al. (2010)
	No ex-type available (Compared with *Xylaria cubensis* GENT 159	*X. cubensis* has smaller conidia size: (3.6–)5–6.3 × 1.8–3.6 μm vs. 4.5–9 × 2.5–4.5 μm	Close to *X. cubensis* GENT 159 (86% ML/0.99BPP). Differ by 1.13% (6/530 bp) of TUB2	Rogers 1984, Rodrigues et al. (1993), Hsieh et al. (2010)
*Xylaria* sp.2 MFLUCC 21-0014	No ex-type available (Compared with *Xylaria phyllocharis* 528 (HAST))	Morphological characters are not sufficient for characterisation	Close to *X. phyllocharis* (75% ML/0.97BPP). Differ by 6.85% (58/847 bp) of ITS; 16.6% (215/1298 bp) of RPB2; 7.17% (38/530 bp) of TUB2	Rodrigues et al. (1993), Hsieh et al. (2010)
*Xylaria* sp.3 MFLUCC 21-0059	*Xylaria bambusicola* WSP 205	Morphological characters are not sufficient for characterisation	Close to *X. bambusicola* (100%ML/1BPP). Differ by 5.31% (45/847 bp) of ITS, 6.04% (32/530 bp) of TUB2 and 4.93% (64/1298 bp) of RPB2	Rodrigues et al. (1993), Hsieh et al. (2010), Dai et al. (2016)

Note: All novel taxa are represented by ex-type.

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
