# Peer review of "Multigene Phylogeny Reveals Endophytic Xylariales Novelties from Dendrobium Species from Southwestern China and Northern Thailand"

_jof, 2022, doi:10.3390/jof8030248_

Round 1

Reviewer 1 Report

Kindly see marginal notes for comments and suggestions.

Author Response

Thank you very much for your valuable comments.

Reviewer 2 Report

In the article entitled “Multigene phylogeny reveals endophytic Xylariales novelties from Dendrobium species from southwestern China and northern Thailand”, Xiao-Ya et.al investigated Xylariales-associated with endangered orchid Dendrobium- identifying several novel Xylariales species as endophytes from Dendrobium using a multigene approach. Authors resolved the taxonomy of novel strains using likelihood and Bayesian approaches. This study is an important contribution to the scientific community for a better understanding of the unknown Xylariales associated with Dendrobium.

I suggest authors rewrite the discussion part to make it reader-friendly. Please see my following suggestions.

Minor comments

Line 29 – Change “Two strains were only identified genus and family ….” to “Two strains were only identified at genus and family….”

Line 1036 – Check the font of species name.

Lines 1043 –Change to “sequence mixed environmental bulk samples…”

Line 1043 – “production of” is not appropriate here. Change “The production of tens of thousands genetic reads…” to “The collection of resulting set of tens of thousands genetic reads….”

Line 1047-1050 – Confusing sentence. Please check if the authors intend to use two sentences. Or consider rephrasing to “The fungal endophyte taxonomic position test of Dendrobium officinale implemented by high-throughput results (HTS) from ITS-rRNA metagenomics analysis showed a very low frequency which perhaps due to a limited number of primer pairs designed for all endophytes [111].

Line 1052 – Please clarify “separated genomic reads”

Lines 1060-1063 – Repetition of lines 1047-1050. Remove the sentence.

Line 1066 – Change “found” to “identified”

Lines 1067-1068 – Check the font.

Lines 1078 – Change to “For further understanding of xylarialean…”

Author Response

(The authors gave the same response as above.)
